# SPARTA: Interpretable functional classification of microbiomes and detection of hidden cumulative effects

**Baptiste Ruiz**[1], **Arnaud Belcour**[1,2], **Samuel Blanquart**[1], **Sylvie Buffet-Bataillon**[3,4], **Isabelle Le Huërou-Luron**[3], **Anne Siegel**[1], **Yann Le Cunff**[1] *

**1** University Rennes, Inria, CNRS, IRISA, Rennes, France, **2** University Grenoble Alpes, Inria, Grenoble, France, **3** Institut NuMeCan, INRAE, INSERM, Univ Rennes, Saint-Gilles, France, **4** Department of Clinical Microbiology, CHU Rennes, Rennes, France

* yann.le-cunff@irisa.fr

**Data Availability Statement:** The source code and data used to produce the results and analyses presented in this manuscript are available on

## Abstract

The composition of the gut microbiota is a known factor in various diseases and has proven to be a strong basis for automatic classification of disease state. A need for a better understanding of microbiota data on the functional scale has since been voiced, as it would enhance these approaches' biological interpretability. In this paper, we have developed a computational pipeline for integrating the functional annotation of the gut microbiota into an automatic classification process and facilitating downstream interpretation of its results. The process takes as input taxonomic composition data, which can be built from 16S or whole genome sequencing, and links each component to its functional annotations through interrogation of the UniProt database. A functional profile of the gut microbiota is built from this basis. Both profiles, microbial and functional, are used to train Random Forest classifiers to discern unhealthy from control samples. SPARTA ensures full reproducibility and exploration of inherent variability by extending state-of-the-art methods in three dimensions: increased number of trained random forests, selection of important variables with an iterative process, repetition of full selection process from different seeds. This process shows that the translation of the microbiota into functional profiles gives non-significantly different performances when compared to microbial profiles on 5 of 6 datasets. This approach's main contribution however stems from its interpretability rather than its performance: through repetition, it also outputs a robust subset of discriminant variables. These selections were shown to be more consistent than those obtained by a state-of-the-art method, and their contents were validated through a manual bibliographic research. The interconnections between selected taxa and functional annotations were also analyzed and revealed that important annotations emerge from the cumulated influence of non-selected taxa.

## Author summary

The field of personalized medicine has major stakes in using an individual's microbiota as a descriptor of health. This raises the question of the interpretability of microbiotal

GitHub at https://github.com/baptisteruiz/SPARTA.git. A supplemental archive with detailed outputs and material for reproduction is available on Zenodo at https://doi.org/10.5281/zenodo.10728697.

**Funding:** The author(s) received no specific funding for this work.

**Competing interests:** The authors have declared that no competing interests exist.

signatures found for various diseases. To gain insight into this matter, we developed the SPARTA (Shifting Paradigms to Annotation Representation from Taxonomy to identify Archetypes) pipeline to highlight and interlink significantly discriminating taxa and metabolic functions. SPARTA relies on the integration of the information from the UniProt database concerning the gut microbiota's functional annotation to microbial abundance data, and Machine Learning classification, with the novel preconception of keeping explicit and thorough information on the connections between taxa and annotations. Through iteration, this method can output a reduced list of the microbiotas' descriptors, both in terms of microbial taxa and functions, with insight into their robustness, for better ease of downstream interpretation. The selection was compared to state-of-the-art approaches, and its contents were validated through a manual bibliographic check of its outputs. Finally, we highlight how discriminant metabolic functions may arise from the aggregation of several low-abundance taxa, giving visibility to these functions which are therefore not easily derivable from approaches based on microbial composition, marking them as potentially novel leads.

## Introduction

The importance and perspectives opened by the human gut microbiota have been at the forefront of the discussion in the medical field in the past years, as a wide array of unsuspected impacts on host health have been derived from its composition. When studying the gut microbiota, the taxonomic scale has generally been favored, to identify biomarkers for various conditions [1–3]. In recent years, however, some voices in the medical community have called for increased inclusion of the gut microbiota's functional paradigm in coming analyses. Specifically, taxonomy-based approaches do not properly account for functional redundancies between species and, in turn, might fall short in identifying novel biochemical pathways that should be targeted by innovative therapies [4].

Functional profilings can be built with several methods, depending on the upstream sequencing method. For raw shotgun metagenomic sequencing (MGS) reads, various tools have been developed for functional analysis, notably including the HuMAnN pipeline [5–7] which can quantify functional annotations (FAs) in a sample based on sequence alignments. For processed 16S sequencing data, PiCRUSt2 [8, 9] stands as one of the most popular tools for functional profiling. Other tools can be agnostic in regard to the sequencing method, such as the EsMeCaTa pipeline [10], which functionally annotates an input list of taxonomic affiliations according to the content of the UniProt database. All of these tools associate FAs to taxa via the interrogation of internal or external databases, creating a link between the taxonomic and functional paradigms.

The resulting functional profiles constitute a basis for uncovering functional markers within the gut microbiota, provided these markers can be ranked or filtered based on how informative they are. Such a ranking can be handled through a linear approach, for example using the limma tool [11], which fits a Generalised Linear Model over the data before testing whether each variable's regression coefficient is significantly different from zero [12–14]. Previous studies in clinical predictive modeling have also highlighted the potential for tree-based methods to perform such a variable selection, such as Random Forests (RFs) [15] thanks to their inherent aptitude for variable ranking through the Gini feature importance metric [16].

RFs are also particularly relevant in this regard, due to their proven efficiency in classifying microbiota data [17], outperforming other classic techniques, such as Support Vector Machines (SVMs) [17–19].

While the shift to functional profiles might lead to a decrease in classification performance, the subsequent analyses based on RF feature importance scores singled out impactful metabolic functions [20, 21]. Obtaining a set of discriminant functions is one of the major aspects when turning to FAs instead of taxa descriptors. However, the usual number of FAs identified in in microbiota data is not always easily tractable (2895 ECs derived from 121 species with HuMAnN3 in context of a meta-analysis of Colorectal Cancer cohorts for example [7]), both for interpretation and for ML algorithms. As a result, the question of variable selection, that is selecting a meaningful subset among all FAs, remains a crucial post-processing step to deliver tractable results.

Variable selection based on ranked features can be established through a fixed criterion. For instance, the limma tool can be coupled with a selection based on adjusted p-value [12–14]. Variable selection can also take the form of a set amount of top features from the list, as implemented by MetAML [19] for example, which searches for the optimal top-k features that maximize classification performance, for k in a set list of values. Iterative approaches, such as the RF-based backward elimination procedure (RVFS) [22], wherein a set fraction of the dataset's variables, chosen at the bottom of the Gini Importance Score's ranking, is iteratively removed until the model reaches peak performance, are also applicable. These methods, however, all require a choice of discrete parameters: RVFS iteratively selects a predetermined percentage of the dataset, and MetAML and limma's approaches cover only an empirically chosen p-value or set of top k values. This advocates the interest of a fully automated selection process, to remove user-induced bias altogether, though the evolution of classification performance should still be controlled.

Another aspect of these selection processes to take into account is that of the variable selection methods' robustness: how does the selected list of features change with slight perturbations in the dataset? One measure of such robustness can be derived from the coherence of repeated selection tasks, using resampling. RF models have been proven to be coherent in the right conditions, but their robustness is also highly dependent on the data and chosen approach [23]. As such, an internal measurement of the RF selections' robustness should be envisaged to add transparency if we are to exploit these selections for downstream biological interpretation. This aspect of the method is evaluated by none of the previously mentioned approaches and remains crucial to ensure robust biological interpretations.

In this article, we present a novel approach, implemented as an automated pipeline named SPARTA (Shifting Paradigms to Annotation Representation from Taxonomy to identify Archetypes). Our method makes it possible not only to exploit the RF as an automated variable selector to improve its performances but also to internally evaluate a variable's robustness as a predictor, for better interpretability of the model. Taking as input taxonomic abundance tables for microbiota samples within a dataset together with health status, SPARTA first retrieves the microbiota's metabolic mechanisms, regardless of the upstream sequencing method. Then SPARTA extracts significantly discriminating features from this process while ensuring consistent classification performances when switching from taxa to FAs as a basis for classification. To achieve that goal, SPARTA extends the MetAML and DeepMicro [18, 19] procedures in three dimensions (increased number of trained random forests, selection of important variables with an iterative process, repetition of full selection process from different seeds) to ensure full reproducibility and exploration of inherent variability in performances due to changes in training hyperparameters.

This approach was tested on six different datasets pre-processed and used as a reference for classification performance by previous works [18, 19]. A post-processing method is also implemented, to accentuate emphasis on genericity and robustness. This involves extracting an adaptive and robust shortlist of significantly discriminant features compiled from a repetition of the method, which we backed through a comparison with selections based on limma and with a manual bibliographic verification. Our pipeline also integrates and exploits the interconnections between organisms and FAs, to also show that cumulative phenomena can be identified by leveraging the relationships between taxa and their expressed FAs.

## Results

### SPARTA overview: Paired mechanistic analysis from relative microbial abundance profiles

SPARTA (see Fig 1) requires two inputs. The first is a table describing the microbial relative abundances (i.e.: taxonomic abundance tables) for each microbiota sample within the dataset, from which functional profiles will be computed. The other is a vector file indicating the groups according to which each sample within the dataset should be classified, represented as green and red colors in Fig 1.

SPARTA is based upon the MetAML and DeepMicro [18, 19] procedures which describe the average results of, respectively, 20 and 5 RFs' training from a predefined seed. To gain robustness, SPARTA trains 20 independent random forests (from a parameterized seed) to predict the patient's status. From the importance score computed on these 20 RFs, SPARTA extracts a shortlist of important features and trains 20 new RFs. This procedure is then repeated on this shortlist until a drop in performance is observed (see Fig 1). This extension of the MetAML and DeepMicro procedures in 3 dimensions (20 random forests, a different seed for each of the 10 runs, and an iterative process to select important variables) is a guarantee for robustness. SPARTA also allows the user to set the seed for each run, ensuring full reproducibility and exploration of inherent variability in performances due to changes in training hyperparameters (see Fig 2).

SPARTA computes three major outputs. The first is a functional table: by using the EsMeCaTa tool [10] to query the UniProt [24] database, we associate a representative proteome to each taxon from the original profiles, and link them to FAs (Gene Ontology (GO) terms [25] and Enzyme Commission (EC) numbers [26]). The prevalence of each of the obtained annotations within the individual samples and abundance data are used to calculate scores of FAs, as described in Materials and Methods. This manipulation ensures that the reference-based method EsMeCaTa provides a quantitative annotation-based description of the gut microbiota.

The second output consists of classification performances: SPARTA, by default, trains RF [16] classifiers on the obtained functional profiles, and measures their performance in categorizing the samples. It also offers the option to train SVM classifiers.

Finally, SPARTA generates a list of features, both taxa and FAs, which are identified as significantly discriminating between the given sample groups based on an automatically calculated selection threshold applied to their average importance scores (Gini or SHAP values, see Materials and methods). The associations between taxa and annotations are also made explicit, allowing each feature to be linked notably to its significant counterparts.

This process generates shortlists of significantly discriminating features that can be combined for a robust consensus. SPARTA is applied 10 times, each time with different test subsets, leading to some differences in variables considered significant. To address this, variables are categorized as follows: **(i)** "Robust" if unanimously deemed significant in all SPARTA runs

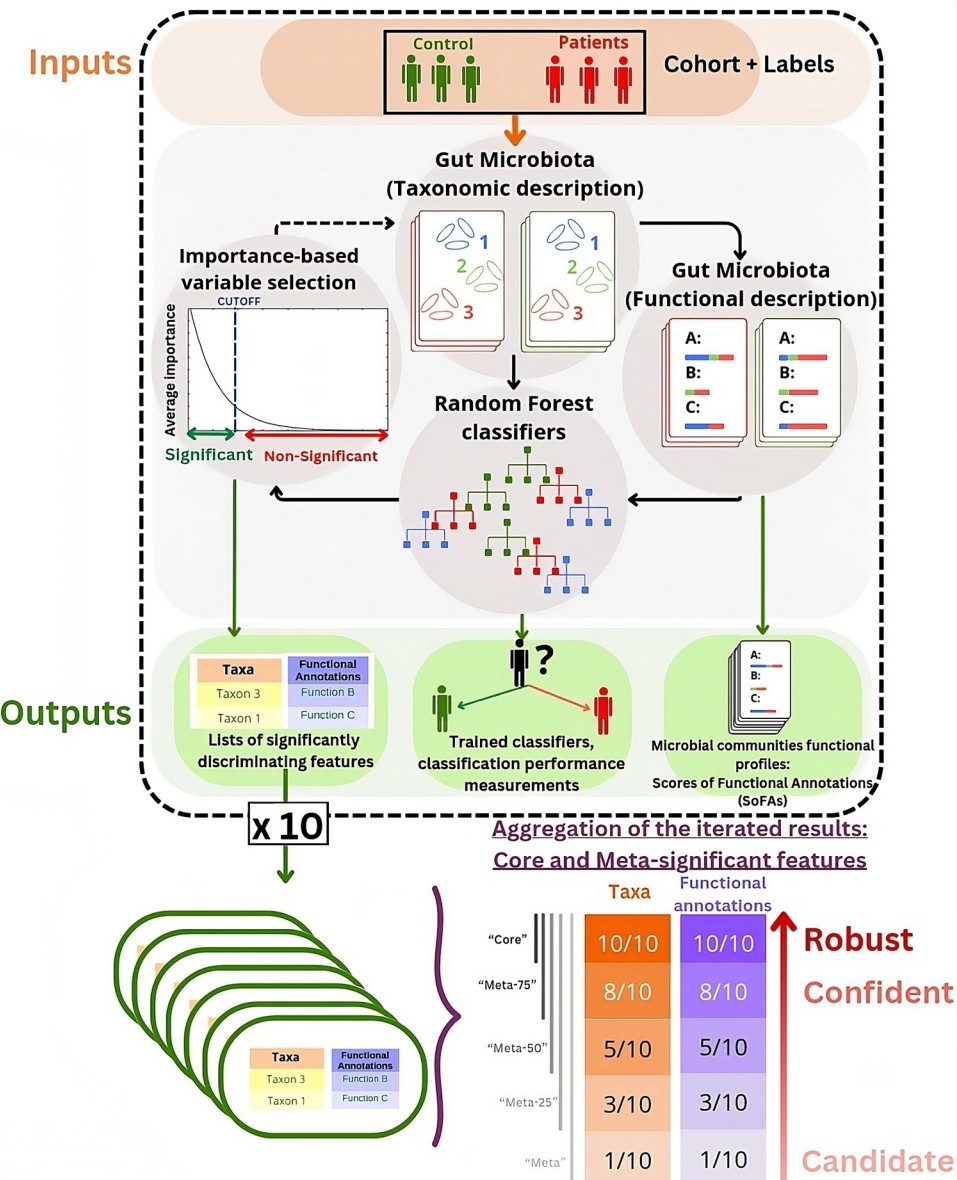

**Fig 1. A schematic representation of SPARTA's pipeline.** From taxonomic tables and their associated labels as inputs, the pipeline produces functional descriptions of the microbiota samples via the EsMeCaTa pipeline. Both of these profiles are then used as basis for the training of RF models to discern Control from Patient profiles. The average importance scores of these variables over all trained forests are then used as basis for a selection of significantly discriminant variables, which can then be processed again iteratively, or passed as an output. For robustness, the process is repeated 10 times, leading to 10 different lists of significantly discriminant taxa and FAs. These lists can be compiled into different categories, which group variables by level of robustness based on the frequency of their appearance in the significant lists. Thus, unanimous variables are considered to be "robust" discriminators, those agreed on by 75% or more of the classifiers are considered "confident", and those that are selected at least once are considered "candidates". Internally to the pipeline implementation, robust features are labeled "Core-significant", and the others are labeled as "Meta-X significant", X being the percentage of significant variable lists that include them.

(above the variable selection threshold). This category contains the variables that are most essential to the discernment of both patient profiles. **(ii)** "Confident" for the variables that were considered significant by at least 75% of the different runs (in our case, by 8 or more runs out of 10). This category contains variables that are likely to be important for profile

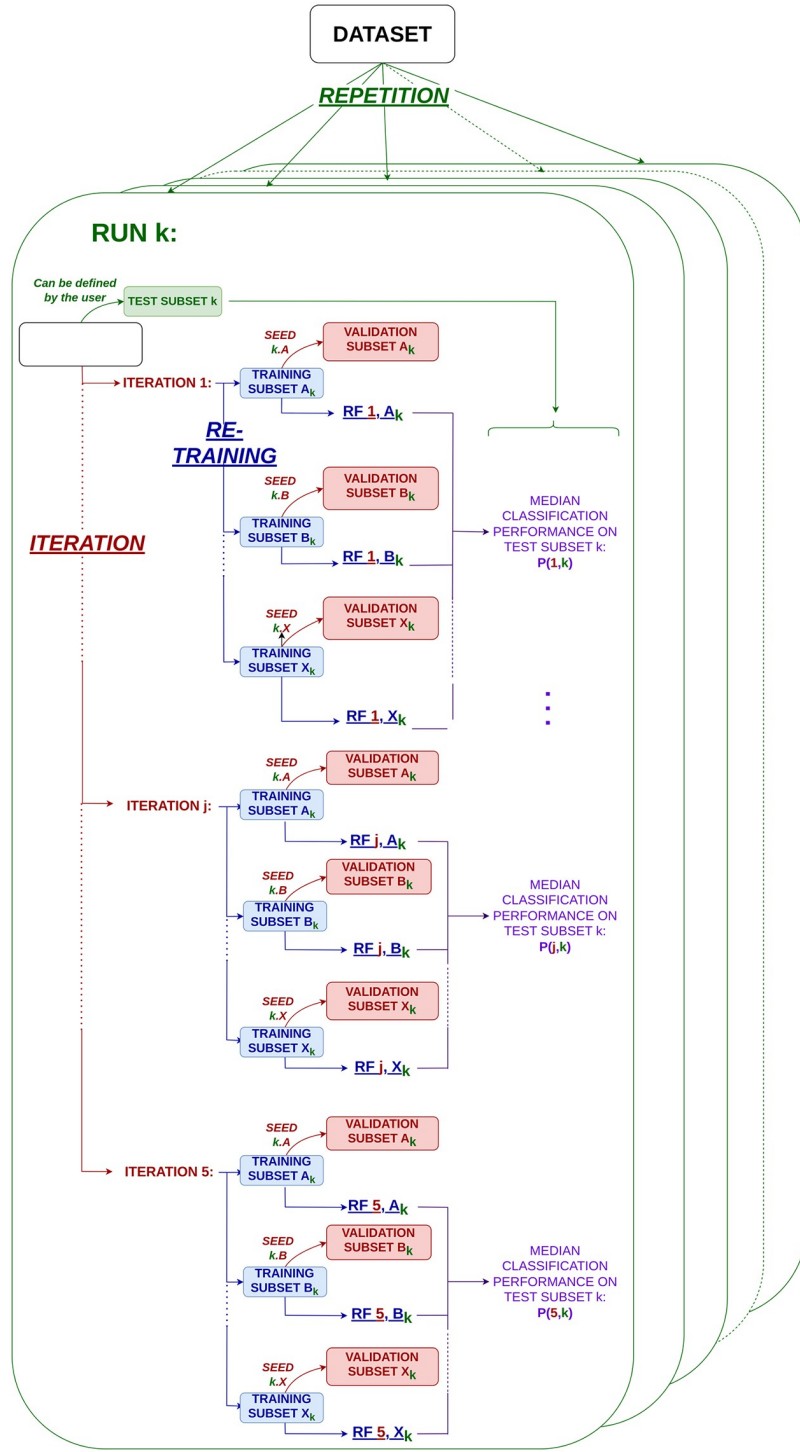

**Fig 2. Classification algorithm implemented in SPARTA.** For a given run *k*, a test subset is randomly selected within the initial dataset and set aside. A given iteration *j* consists in training X random forests (20 by default), each having a dedicated validation subset. These 20 forests are used to compute a median classification performance $P(j, k)$ and a shortlist of important features. This lists is used to train the X random forests of iteration $j + 1$. By default, SPARTA launches 10 runs and 5 iterations.

discrimination and could be a complement to the robust shortlist for interpretation. **(iii)** "Candidate" for variables shortlisted in at least one SPARTA run. These are variables that should not be fully excluded from consideration when it comes to interpretation, but that are unlikely to be influential. More generally, across all of these categories, the robustness of a selected variable can be evaluated in light of the number of different SPARTA runs that list it as significantly discriminant.

Overall, taxa and FAs are quantified on three different levels by SPARTA. They are given: **(i)** A score based on their presence in each individual sample, in the form of a matrix containing, per sample, the relative abundances for taxa, or the scores for annotations (output 'SoFA_table.tsv'), **(ii)** A quantification of how discriminant they are between profiles of samples in the form of a vector of importance scores, **(iii)** An indicator of their robustness as a discriminator, in the form of lists of variables affiliated to the "robust" and "candidate" categories.

## Differential analysis of taxonomic and functional results

**Experimentation.**   We applied SPARTA to six publicly available datasets, previously explored in articles such as MetAML [19] or DeepMicro [18]. These datasets contain taxonomic abundance tables issued from sequenced microbiota samples from cohorts of healthy controls and individuals diagnosed with Cirrhosis (Cirrhosis dataset), Colorectal Cancer (Colorectal dataset), Obesity (Obesity dataset), Type 2 Diabetes (T2D and WT2D datasets) or Inflammatory Bowel Disease (IBD dataset). For further details, see Materials and methods. SPARTA was launched with both RFs and Gini importance score [16] (default parameters), and with RFs and SHAP [27] importance score, for 10 runs, 5 selection iterations per run, and 20 trained models per iteration. We also evaluated the classification performances of SVM classifiers on the same datasets.

**Machine Learning classification performances obtained from functional profiles are similar to those from taxonomic profiles.**   Fig 3 illustrates the classification performances of the RFs [16] trained by SPARTA to distinguish between patients and healthy individuals, per profile and dataset. Classification on the taxonomic datasets prior to selection is analogous to the classification without representation learning method implemented in DeepMicro [18], with 20 RF (SPARTA) instead of 5 (DeepMicro) and dedicated test sets (SPARTA). For each RF trained by SPARTA, the area under the receiver operating characteristic curve (AUC) is calculated. Seeing as 20 RFs are trained within an iteration, the median of these 20 AUCs is retained to represent the performances of the iteration as a whole. The full iterative process is repeated 10 times, giving 10 median performance metrics per level of iterative selection (see Fig 2). The amount of selections that leads to the highest median among these metrics is deemed to be the optimal selection and is the one represented here for the taxonomic and functional profiles. The number of selective iterations corresponding to this selection are given in the 'Optimal Selection' column. For each dataset, a Mann-Whitney U-test was conducted comparing the performances based on the taxonomic and functional profiles at respective optimal selection levels. The details of all of the obtained performances are given in S1 File. We also performed the same process evaluation with SVMs and showed that RFs consistently outperform SVMs on the datasets presented in this paper (see S1 Fig), as well as RF with SHAP values as importance scores (see S1 Fig, no significant difference compared to RF and Gini importance scores). We also reported in S1 Table the classification performances on validation sets. Those are usually higher than on the test set. This is expected due to the iteration process where importance scores are averaged over all RF to perform variable selection.

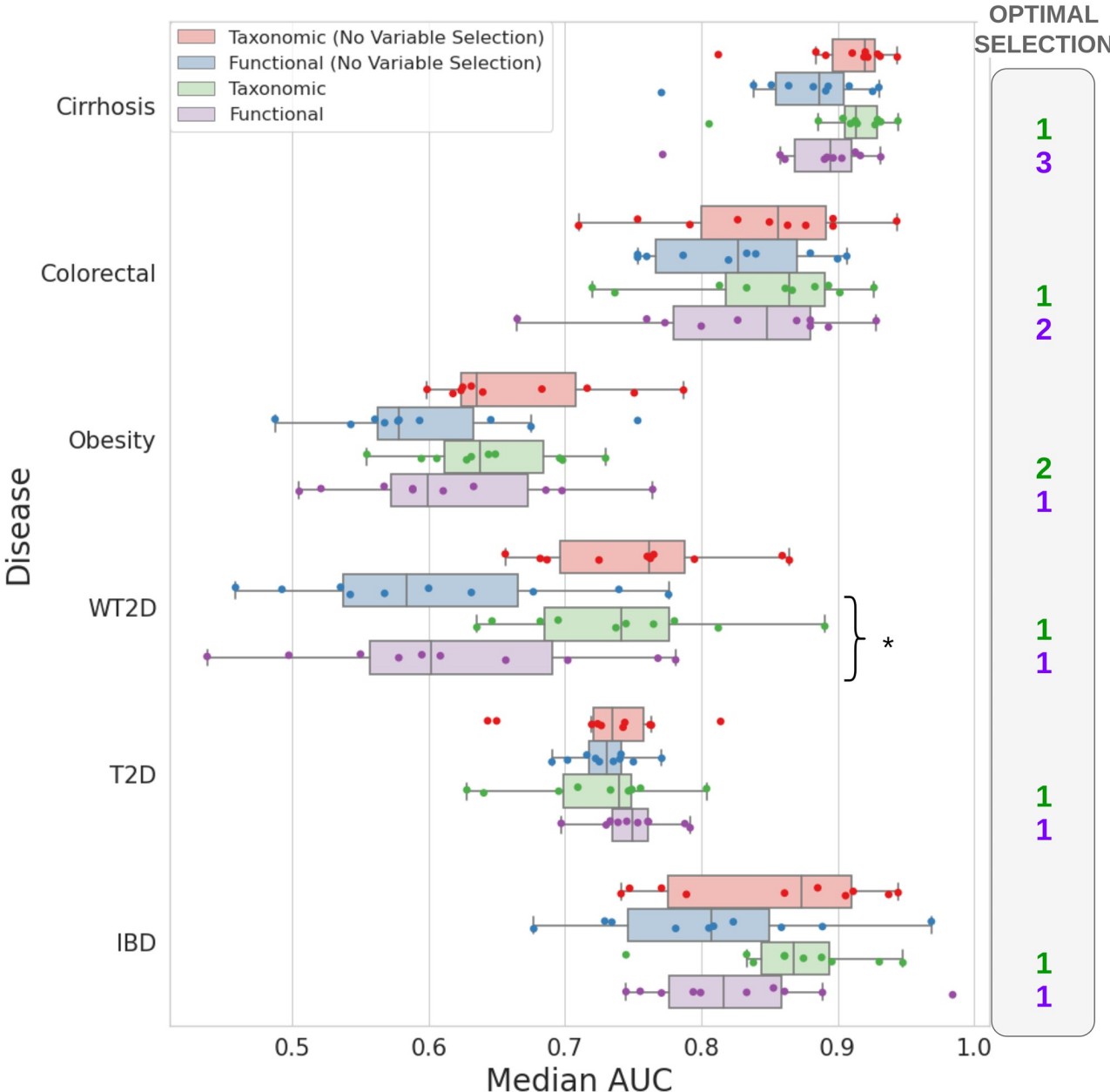

**Fig 3. Classification performances of RF models trained on taxonomic and functional profiles, and impact of the variable selection on performance.** Median classification performances (AUC) for all types of profiles and each dataset, on the original datasets as well as at the optimal level of selection over 10 full runs of the pipeline. Each of these runs involved a different randomly selected test set of individuals, which was used for both profiles. Performances and importance scores for each run were computed and averaged over 20 distinctly trained RF models. The amount of selection iterations required to obtain the best average among these median AUCs are represented beside each plot. Instances when the difference in performance between functional and taxonomic profiles using SPARTA is significant for a same dataset (based on a Mann-Whitney U-test) are signaled by a * symbol.

For example, the Colorectal dataset's functional (purple) and taxonomic (green) profiles have been tested over 10 runs by SPARTA. These tests have allowed us to detect the level of variable selection that yields the best median classification scores for each profile, which were then chosen for this representation. In this case, as shown in the 'Optimal selection' column,

the functional dataset gives its best performance after 2 iterations of variable selection, whereas the taxonomic dataset gives its best performance after just one. The performances of RFs trained on taxonomic and functional profiles without selection are also represented, in red and blue respectively. Each of the 10 runs of SPARTA yields an average classification performance score, corresponding to the plotted dots. The boxplots represent the associated distribution and notably show that the functional profile has a median AUC of 0.85, against 0.86 for the taxonomic profile. The difference between both distributions was not found to be significant by a Mann-Whitney U-test, as shown by the absence of an asterisk symbol on this row.

Overall, we can see that taxonomic profiles yield better median classification performances than their functional counterparts, with the T2D dataset being the only exception. However, the difference in performance between both profiles is only significant in the case of the WT2D dataset, showing that though converting our data to the functional level comes at the cost of some performance, both profiles perform comparably as a basis for classification. These results are in line with the previous works of Douglas et al. [20] and Jones et al. [21]. However, the innovative potential of functional profiles resides more in their prospective contribution to a biological understanding of the diseases' mechanisms than in their use for automatic classification.

Of note is also the asymmetrical benefit of variable selection. Functional profiles systematically benefit from a reduction of dimensionality, as their median performances after iterative selection (purple) are always superior to those obtained without variable selection (blue). For taxonomic profiles however, variable selection leads to a decrease in median results for three of the six datasets (Cirrhosis, WT2D, and IBD).

A comparative classification was made based on a functional profile built from the raw reads of the IBD dataset with HuMAnN3 [7], using the same parameters. The obtained results (S2 Fig) show that median classification based on functional profiles built directly from the reads are on par with those obtained using EsMeCaTa, as the differences in performance are not significant based on a Mann-Whitney U-test (p-value = 0.45). Both functional profiles' performances are also non significantly different from the performance obtained on the IBD taxonomic dataset (p-value = 0.73 for HuMAnN and 0.36 for EsMeCaTa).

We will now focus on propositions to optimize the differential functional profiling of microbiotas in the context of a disease, as well as evaluate the added value of functional information in comparison to taxa for understanding the underlying biological processes.

**Robustness of SPARTA's feature selection: Comparative evaluation.** The datasets used in the previous section contained on average 484 taxa. Through EsMeCaTa's [10] pipeline and its interrogation of UniProt [24], these taxa were linked to a total average of 10,510 FAs per dataset, resulting in a 22-fold mean increase in the amount of information, as shown in Table 1. For example: in total, the sequenced samples of the Cirrhosis dataset covered 542 taxa, which were associated by EsMeCaTa to a total of 10,434 FAs. Following SPARTA's application, 72 of these taxa and 33 of these annotations were included in the candidate sublists. Among these, 32 taxa and 7 annotations were in the confident subset, and 23 taxa and 4 annotations were in the robust subset. The sizes of the selections obtained from selections based on SHAP importances are also available in S3 and S4 Figs. These results showcased that selections based on SHAP importance scores were less robust than those based on Gini importance, as the sizes of the Robust selections obtained through this method were consistently smaller than those obtained with Gini-based selection, for both functional and taxonomic profiles. Three of the functional datasets (WT2D, Obesity, and Colorectal) and one functional dataset (Colorectal) even gave empty Robust selections from the first iteration with SHAP, which does not happen on any dataset with Gini. As such, only results based on Gini selections were presented here. This is however illustrative of the impact of the ranking approach on the overall quality of SPARTA's analysis.

**Table 1. Application of the SPARTA selection process to identify signature taxa and functions on 6 reference datasets.**

| Dataset | Features | Initial number of taxa | Predicted Functions | Robust subset | Confident subset | Candidate subset |
|---|---|---|---|---|---|---|
| Cirrhosis | Taxa | 542 | - | 23 | 32 | 72 |
|  | FAs | - | 10,434 | 4 | 7 | 33 |
| Colorectal | Taxa | 503 | - | 24 | 37 | 109 |
|  | FAs | - | 10,635 | 1 | 17 | 355 |
| Obesity | Taxa | 465 | - | 136 | 154 | 188 |
|  | FAs | - | 11,341 | 26 | 169 | 3,199 |
| WT2D | Taxa | 381 | - | 27 | 51 | 136 |
|  | FAs | - | 10,180 | 8 | 69 | 3,150 |
| T2D | Taxa | 572 | - | 117 | 136 | 202 |
|  | FAs | - | 10,275 | 139 | 307 | 1,575 |
| IBD | Taxa | 443 | - | 22 | 29 | 100 |
|  | FAs | - | 10,196 | 59 | 167 | 1,883 |

Total amount of features (taxa and FAs) in the original dataset ("Initial Number" column) and in the robust, confident, and candidate selections at the optimal SPARTA selection threshold (Calculated over 10 runs of the pipeline).

To balance the increase in information when using FAs, SPARTA operates a selection of variables based on the features' importance scores. These scores, when ordered from highest to lowest, display a kink-like shape. SPARTA automatically operates a cut-off at the inflection point of the kink and probes whether classification performances are improved (see Materials and methods). This selection aims to correct the redundancies and the dimensionality of the original dataset for better classification. It also generates one of the pipeline's main outputs: a list of ranked features (either taxa or FAs) based on their average importance scores [16], and including an automatically computed cutoff that distinguishes significant and non-significant information. SPARTA provides the user with the list of important taxa and FAs for each iteration, the corresponding classification performance, and a focus on the best iteration after the first level of selection.

The amount of information retained per SPARTA run for all functional datasets is illustrated in Fig 4(A). The figure shows that the average amount of information to retain for optimal classification performance varies depending on the dataset. For example, retaining the top 500 annotations ranked by average Gini importance would give a selection similar to SPARTA on the IBD dataset, whereas the Obesity dataset would require the top 1,000 annotations to match the selection. This shows that an adaptive method like SPARTA, which makes a decision concerning the quantity of information to be retained by the selection, has an advantage over a selection based on a fixed threshold because it can adapt to the complexity of the problem at hand, which is shown here to be variable. SPARTA's selection thresholds also do not match the more traditional thresholds, such as the top 30 features explored in Jones et al. [21], and can be used to get an estimate of the optimal amount of information to consider for discerning microbiota profiles.

To explore whether SPARTA's variable selection differs from classic linear approaches, we compared our approach with a standard method designed for continuous data [11] rather than for count data [28]. Specifically, selections obtained from direct pairwise comparison of the profiles using the limma tool [11] were compared. Variables were selected using a p-value threshold of 0.05, a classic threshold value exploited in several other studies that applied limma to metagenomic data [12–14]. Similarly to SPARTA, the selection process was iterated 10

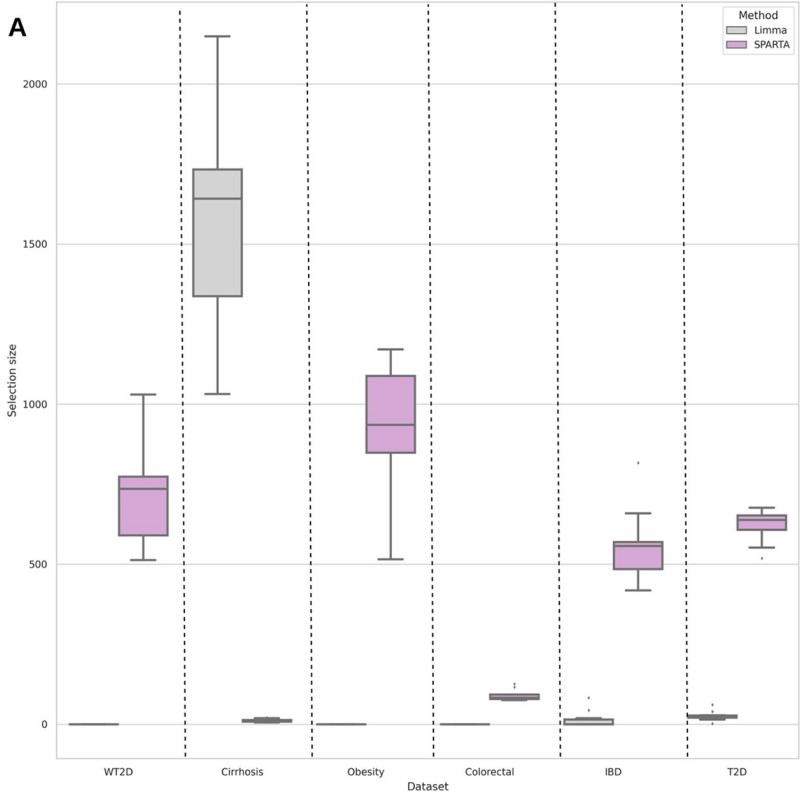

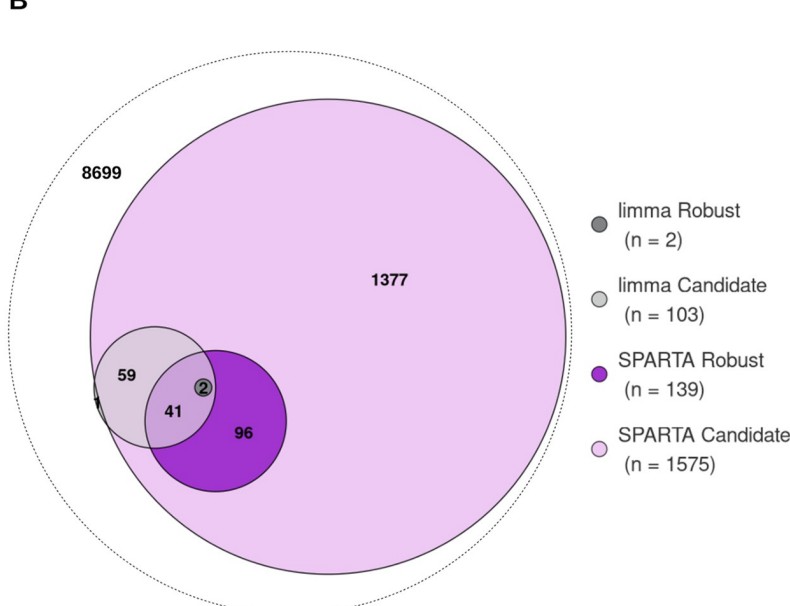

**Fig 4. Comparison between SPARTA and limma functional selections. A: Number of important selected FAs for each run at best iteration for the six datasets** Amount of FAs selected by SPARTA and limma, for all datasets. Limma selections were effectuated with an adjusted p-value threshold of 0.05. Both selection methods were repeated 10 times, with a common test subset set aside each time. **B: Comparison between robust and candidate FAs for T2D dataset** The limma subsets were obtained using the classic threshold of 0.05. Values indicate the number of annotations in each intersection and do not represent the size of a category as a whole. The white circle includes all annotations from the full dataset.

**Table 2. Sizes of the SPARTA and limma selections.** Limma was applied with an adjusted p-value threshold of 0.05. From left to right, the columns present, for SPARTA and limma, the size of the robust, confident, and candidate subsets issued by the concerned selection method iterated 10 times with identical test subsets.

| | Total size of the robust subset | | Total size of the confident subset | | Total size of the candidate subset | |
|---|---|---|---|---|---|---|
| | **SPARTA** | **Limma** | **SPARTA** | **Limma** | **SPARTA** | **Limma** |
| Cirrhosis | 4 | 878 | 7 | 1,165 | 33 | 2,668 |
| Colorectal | 1 | 0 | 17 | 0 | 355 | 0 |
| Obesity | 26 | 0 | 169 | 0 | 3,199 | 0 |
| WT2D | 8 | 0 | 69 | 0 | 3,150 | 0 |
| T2D | 139 | 2 | 307 | 4 | 1,575 | 103 |
| IBD | 59 | 0 | 167 | 0 | 1,883 | 111 |

times with variation induced from setting aside a subset of the samples, and variables were compiled into 'robust', 'confident', and 'candidate' categories depending on how often they were selected. Comparative results of this process are presented in Fig 4(A) and Table 2. For example, Fig 4(A) shows that, when applied 10 times to the Cirrhosis dataset, SPARTA selects a minimum of 6 annotations, and a maximum of 21, with a median of 11. In the same conditions, limma selects between 1,032 and 2,149 annotations, for a median of 1,642. These distributions are plotted, respectively, in purple and gray. Table 2 shows that with SPARTA's selection, the Cirrhosis dataset outputs 4 robust annotations, 7 confidents, and 33 candidates, against a respective 878, 1,165 and 2,668 with limma. With these parameters, limma is a much more stringent selector than SPARTA on all datasets aside from Cirrhosis. For the Colorectal, WT2D and Obesity datasets in particular, all selections are empty, leading to an empty candidate subset as described in Table 2. The IBD dataset also proves to be unsuitable for this approach, yielding empty robust and candidate subsets, and an empty robust subset. Only the T2D and Cirrhosis datasets allow limma to yield a non-empty robust subset. SPARTA, on the other hand, consistently yields non-empty robust and confident selections, both of which are reasonably sized for interpretation when compared to the candidate subsets, being close to 50 times smaller in the case of the WT2D dataset's confident and candidate subsets.

Among these datasets, Cirrhosis stands out as an outlier. Indeed, it is by far the dataset on which limma selects the most information: in Fig 4(A), we can see that it selects 1550 annotations on average over 10 iterations, whereas the second highest amount, obtained with the T2D dataset, is only 26.1 on average. This also makes it the only case in which SPARTA proves to be the most stringent of the two selectors, with an average of 12 selections per run, for a Robust selection of size 4 against limma's 878 (see Table 2). The four annotations in question are: GO:0016984 (ribulose-bisphosphate carboxylase activity), GO:0003779 (actin binding), GO:0004081 (bis(5'-nucleosyl)-tetraphosphatase (asymmetrical) activity) and GO:0018112 (proline racemase activity). Actin binding (GO:0003779) signals the participation of the gut in the maintenance of the intestinal epithelia, which plays a role in the prevention of liver diseases such as Cirrhosis [29]. The activity of proline racemase (GO:0018112) is also indicative of proline metabolism in the gut, which has also been shown to be upregulated in cases of Cirrhosis [30]. The activity of the bis(5'-nucleosyl)-tetraphosphatase enzyme (GO:0004081) is involved in the metabolism of both purine and pyrimidine according to KEGG [31], which are disturbed in mice gut during the development of Cirrhosis [32]. Finally, ribulose-bisphosphate carboxylase (GO:0016984), though it is mostly known for its role in photosynthesis, can also be involved in the salvage of methionine [33], itself key in the development of liver disease [34].

As such, in the case of Cirrhosis, SPARTA robustly highlights a small subsection of biologically relevant annotations, themselves consistently highlighted by limma as linear indicators of the prevalence of the disease. This could illustrate a case in which the dataset is "too easy" to

predict, due to an abundance of features that linearly differentiate the profiles, and a small sample of which is sufficient to be efficient in classification. This could lead to an over-selection from SPARTA, as even when relevant features are removed by the iterated selection, the remaining variables still allow for good classification performance. In this case, it could be interesting to look at the selections from iterations before the optimum.

We then focused on the T2D dataset, which is the only other dataset on which limma i) extracts a non-empty robust selection with an adjusted p-value threshold of 0.05 (see Table 2) and ii) consistently provides non-empty FA selections. Fig 4(B) illustrates the overlap between limma's and SPARTA's robust and candidate annotations.

T2D's limma selection is smaller than SPARTA's, englobing a total of 103 annotations in its candidate subset against 1,575 for SPARTA, as shown in Table 2. As shown by Fig 4(B), all of these annotations aside from one are included in SPARTA's candidate selection. Similarly, limma's robust subset is entirely included in SPARTA's robust selection.

To put these results in perspective, there is no guarantee that a 0.05 p-value threshold yields an 'optimal' selection for this dataset when applying limma. This choice of threshold is, however, a required external input for the method, that SPARTA does not need as it automates the choice of the selection's size. As such, the chosen threshold could arguably be too restrictive for the T2D dataset. As an illustration, a p-value threshold of 0.255, obtained to generate a limma candidate selection as close as possible to the size of SPARTA's, was applied, as illustrated by S5 Fig. This much less restrictive threshold yields a limma selection that still largely overlaps with SPARTA's selection (74% of limma's annotations being included in SPARTA's).

Overall, limma does not yield exploitable selections with a classic p-value threshold on four of our six datasets. The examination of the remaining two datasets allows us to illustrate how SPARTA and limma behave comparatively in different situations. In T2D's situation, the limma selection is smaller and largely overlaps SPARTA, with limma's robust subset notably being entirely included in the SPARTA selection. For the Cirrhosis dataset, the SPARTA selection is the smallest of the two, however, it remains coherent with what limma selects, and yields information that is coherent with the biological question at hand. RF classification performances obtained on both selections and presented in S6 Fig, also show that limma's selections perform under SPARTA's as basis for classification, as neither of the recorded performances surpass their SPARTA counterparts.

## Exploiting biological knowledge from the paired robust functions and taxa

For the following section, we will be relying on the robust outputs from the IBD dataset as an example. These results come from the pipeline's first iteration, which are the best performing selective iterations for both profiles (see Fig 3). The IBD dataset was chosen as an illustrative representative of our results, as it is an outlier in neither classification performance, being the third best-performing dataset out of six, nor in the selection of variables by limma.

**Visualization of the robust shortlists.** An important output of SPARTA is the shortlist of robust variables that are selected by the method, allowing for downstream interpretability. This comes in the form of tables of robustly significant annotations and taxa, as previously described. The annotation shortlist for the IBD dataset is given in Table 3. It contains 59 FAs, alongside extra information that SPARTA helps associate with them. For example, annotation GO:0006520, corresponding to the amino acid metabolic process, is first in the table because it has the highest average Gini importance score over all 200 forests trained at this selection level, over 10 runs. It is on average 1.05 times as present in the diseased profiles as it is in the controls, the negative value of the 'Ponderated average ratio' meaning that the annotation is predominantly found in unhealthy samples. It is linked to a total of 358 taxa over all samples, of

**Table 3. Robust subset of annotations from the IBD dataset.**

| ID | Names | Average RF importance | Ponderated average ratio (Control/Unhealthy) | Number of linked taxa | | Bibliographic category |
|---|---|---|---|---|---|---|
| | | | | Total | Robust | |
| GO:0006520 | amino acid metabolic process | 4.37E-03 | -1.04801771712759 | 358 | 20 | 1 |
| 4.1.2.- | Aldehyde Lyases | 4.01E-03 | -1.83063815612961 | 28 | 2 | 2 |
| GO:0102545 | phosphatidyl phospholipase B activity | 3.53E-03 | -4.59984365662854 | 15 | 1 | 1 |
| GO:0004122 | cystathionine beta-synthase activity | 3.42E-03 | -3.75076174596704 | 8 | 1 | 1 |
| GO:0008744 | L-xylulokinase activity | 3.24E-03 | -5.44424049313829 | 4 | 1 | 3 |
| GO:0047419 | N-acetylgalactosamine-6-phosphate deacetylase activity | 2.57E-03 | -1.19907351364782 | 78 | 4 | 1 |
| GO:0008788 | alpha,alpha-phosphotrehalase activity | 2.44E-03 | -2.30364417355582 | 19 | 1 | 3 |
| GO:0032440 | 2-alkenal reductase [NAD(P)+] activity | 2.43E-03 | 3.17446593793351 | 5 | 1 | 3 |
| GO:0001510 | RNA methylation | 2.40E-03 | 1.05457228463169 | 249 | 17 | 1 |
| GO:0015444 | P-type magnesium transporter activity | 2.34E-03 | -1.65841492481138 | 66 | 2 | 2 |
| GO:0016832 | aldehyde-lyase activity | 2.24E-03 | -1.12570888096648 | 200 | 12 | 2 |
| GO:0047605 | acetolactate decarboxylase activity | 2.23E-03 | -1.56525594318597 | 48 | 1 | 3 |
| GO:1901135 | carbohydrate derivative metabolic process | 2.18E-03 | -1.10067892552244 | 271 | 14 | 1 |
| GO:0017065 | single-strand selective uracil DNA N-glycosylase activity | 2.14E-03 | 3.15494616303483 | 4 | 1 | 1 |
| GO:0009346 | ATP-independent citrate lyase complex | 2.10E-03 | -1.6037562809426 | 52 | 1 | 2 |
| GO:0016811 | hydrolase activity, acting on carbon-nitrogen (but not peptide) bonds, in linear amides | 2.05E-03 | -1.25265006865793 | 162 | 4 | 1 |
| GO:0008815 | citrate (pro-3S)-lyase activity | 2.03E-03 | -1.63749881151623 | 53 | 1 | 1 |
| GO:0042121 | alginic acid biosynthetic process | 1.94E-03 | 1.14990181028563 | 127 | 12 | 1 |
| 4.1.3.6 | citrate (pro-3S)-lyase. | 1.94E-03 | -1.60221767316624 | 52 | 1 | 1 |
| GO:0047395 | glycerophosphoinositol glycerophosphodiesterase activity | 1.93E-03 | -5.70423027266411 | 2 | 1 | 1 |
| GO:0008092 | cytoskeletal protein binding | 1.90E-03 | 3.03923451098608 | 3 | 1 | 1 |
| GO:0045151 | acetoin biosynthetic process | 1.90E-03 | -1.56525594318597 | 48 | 1 | 3 |
| 4.1.1.5 | acetolactate decarboxylase. | 1.85E-03 | -1.56525594318597 | 48 | 1 | 3 |
| GO:0033711 | 4-phosphoerythronate dehydrogenase activity | 1.79E-03 | 1.21622800529026 | 99 | 6 | 3 |
| GO:0043130 | ubiquitin binding | 1.79E-03 | 2.84452978123873 | 6 | 1 | 1 |
| 2.8.3.10 | citrate CoA-transferase. | 1.78E-03 | -1.57616213702899 | 52 | 1 | 1 |
| GO:0008910 | kanamycin kinase activity | 1.78E-03 | -1.58851951224173 | 11 | 1 | 1 |
| GO:0046537 | 2,3-bisphosphoglycerate-independent phosphoglycerate mutase activity | 1.78E-03 | 1.07286170037927 | 185 | 17 | 3 |
| GO:0047356 | CDP-ribitol ribitolphosphotransferase activity | 1.72E-03 | -6.7139421245469 | 1 | 1 | 2 |
| GO:0000310 | xanthine phosphoribosyltransferase activity | 1.69E-03 | -1.08340423243633 | 201 | 10 | 3 |
| GO:0008814 | citrate CoA-transferase activity | 1.68E-03 | -1.57775123389852 | 52 | 1 | 1 |
| GO:0005727 | extrachromosomal circular DNA | 1.68E-03 | -1.83727037420844 | 13 | 0 | 1 |
| GO:0004792 | thiosulfate sulfurtransferase activity | 1.67E-03 | -1.17227801781067 | 82 | 2 | 1 |
| GO:0008707 | 4-phytase activity | 1.67E-03 | 3.14255076857911 | 1 | 1 | 3 |
| GO:0019677 | NAD catabolic process | 1.64E-03 | 1.30706123906711 | 32 | 1 | 1 |
| GO:0008610 | lipid biosynthetic process | 1.64E-03 | -1.47728727133267 | 87 | 2 | 1 |
| 2.4.2.22 | xanthine phosphoribosyltransferase. | 1.64E-03 | -1.08534297116337 | 199 | 10 | 3 |
| GO:0047330 | polyphosphate-glucose phosphotransferase activity | 1.59E-03 | -3.53503053492603 | 5 | 1 | 1 |
| 2.7.1.23 | NAD(+) kinase. | 1.56E-03 | -1.04348206094609 | 325 | 16 | 1 |
| GO:0016746 | acyltransferase activity | 1.54E-03 | 1.10289112410377 | 347 | 18 | 2 |
| GO:0071702 | obsolete organic substance transport | 1.54E-03 | -1.20335282764238 | 103 | 4 | 3 |
| GO:0006741 | NADP biosynthetic process | 1.53E-03 | -1.04692108483062 | 329 | 16 | 1 |
| 4.2.1.- | Hydro-Lyases | 1.52E-03 | -1.90522797851666 | 19 | 1 | 2 |

*(Continued)*

**Table 3.** (Continued)

| ID | Names | Average RF importance | Ponderated average ratio (Control/Unhealthy) | Number of linked taxa | | Bibliographic category |
|---|---|---|---|---|---|---|
| | | | | Total | Robust | |
| GO:0006144 | purine nucleobase metabolic process | 1.45E-03 | -2.26707747018229 | 21 | 0 | 1 |
| GO:0004135 | amylo-alpha-1,6-glucosidase activity | 1.45E-03 | 1.16316684039 | 73 | 7 | 3 |
| GO:0032265 | XMP salvage | 1.40E-03 | -1.08523959035346 | 199 | 10 | 2 |
| GO:0008760 | UDP-N-acetylglucosamine 1-carboxyvinyltransferase activity | 1.40E-03 | -1.06687124783542 | 361 | 17 | 3 |
| 2.1.1.195 | cobalt-precorrin-5B (C(1))-methyltransferase. | 1.33E-03 | -1.11006222435079 | 89 | 5 | 4 |
| 3.5.3.6 | arginine deiminase. | 1.32E-03 | -1.3947618868725 | 58 | 2 | 1 |
| GO:0003953 | NAD + nucleosidase activity | 1.31E-03 | 1.31930434489008 | 29 | 1 | 1 |
| 1.1.1.22 | UDP-glucose 6-dehydrogenase. | 1.30E-03 | 1.14909369105086 | 142 | 10 | 1 |
| GO:0097056 | obsolete selenocysteinyl-tRNA(Sec) biosynthetic process | 1.29E-03 | -1.23940120402784 | 214 | 5 | 1 |
| GO:0016297 | fatty acyl-[ACP] hydrolase activity | 1.28E-03 | 1.09879874319015 | 122 | 11 | 3 |
| GO:0006522 | alanine metabolic process | 1.24E-03 | -2.03436740514923 | 17 | 1 | 1 |
| GO:0008808 | cardiolipin synthase activity | 1.18E-03 | 1.0806848147406 | 239 | 15 | 3 |
| GO:0009409 | response to cold | 1.13E-03 | -1.9117942663824 | 35 | 0 | 1 |
| GO:0008899 | homoserine O-succinyltransferase activity | 9.43E-04 | -1.07852816252639 | 194 | 11 | 3 |
| GO:0008276 | protein methyltransferase activity | 8.62E-04 | -1.0433934172264 | 283 | 16 | 2 |
| 1.1.1.88 | hydroxymethylglutaryl-CoA reductase. | 6.52E-04 | -1.48308851426957 | 50 | 0 | 1 |

Robust FAs of the IBD dataset, identified by their GO term or EC number, as well as their current name. Annotations are classified by decreasing average Gini importance score, over all 200 RFs trained at the optimal selection level (20 per run, 10 runs). Extra information include: the ratio between the average scores of the annotation in control and unhealthy profiles, ponderated by -1 if the annotation is most present in the unhealthy profiles, the amount of taxa attached to each FA, and the amount of robust taxa within them. Finally, the bibliographic category of each annotation, as defined in a subsequent section, is given.

which 20 were found to be robust. The subsequent bibliographic analysis of this list graded its relevance to the disease as a 1, meaning that there is a known direct link between the annotation and IBD [35]. Detailed outputs are made available in S2 File.

A similar selection of robustly discriminant taxa is also available as an output of the pipeline, with the IBD output given as an example in Table 4. The same information as the previous table is available for each taxon, aside from the bibliographic categories. For instance, *Alistipes finegoldii*, identified in our process as Organism 73, similarly ranks first because it has the highest Gini importance score on average over all trained RFs. Its differential expression shows that it is expressed on average 16 times as much in control profiles as it is in the unhealthy samples. As previously, we can establish which annotations are attached to each taxon, with *A.finegoldii* expressing a total 1,220 FAs, 15 of which are robustly significant. The details of these associations are available in S3 File.

**Bibliographic exploration of the functional robust shortlist.** Beyond the examples mentioned in this chapter, an in-depth bibliographic analysis of these outputs has been conducted for the IBD dataset and is available in S4 File.

The bibliographic examination was conducted on the integrality of the robust annotations from the IBD dataset, as well as samples of 20 annotations that were present in 50% of the significant sublists from SPARTA's runs, and 20 non-candidate annotations. The methodology was to research the name of the annotation alongside the name of the disease on Google Scholar (https://scholar.google.com/). If none of the research results provided conclusive information linking this annotation to IBD, be it in a host model or the microbiota, the

**Table 4. Robust subset of taxa from the IBD dataset.**

| ID | Names | Average RF importance | Ponderated average ratio (Control/Unhealthy) | Number of linked annotations | |
|---|---|---|---|---|---|
| | | | | Total | Robust |
| Organism_73 | Alistipes finegoldii | 2.84E-02 | 16.3964097691144 | 1220 | 15 |
| Organism_224 | Akkermansia muciniphila | 2.12E-02 | 3.23501956449667 | 1452 | 18 |
| Organism_12 | Bifidobacterium bifidum | 2.03E-02 | -11.3214966525224 | 1307 | 28 |
| Organism_144 | Lachnospiraceae bacterium 2 1 58FAA | 1.91E-02 | -18.4267002012075 | 355 | 5 |
| Organism_169 | Ruminococcus lactaris | 1.90E-02 | 3.04069705100761 | 431 | 5 |
| Organism_127 | Beubacterium ventriosum | 1.51E-02 | 2.6429359268965 | 1388 | 20 |
| Organism_156 | Oscillibacter unclassified | 1.42E-02 | -1.89778568117644 | 724 | 16 |
| Organism_134 | Butyrivibrio unclassified | 1.39E-02 | -1.65319948992604 | 916 | 10 |
| Organism_54 | Odoribacter splanchnicus | 1.33E-02 | 1.85573337062113 | 1595 | 19 |
| Organism_75 | Alistipes onderdonkii | 1.33E-02 | 2.48594496944176 | 1391 | 15 |
| Organism_78 | Alistipes shahii | 1.30E-02 | 1.65146163415684 | 935 | 8 |
| Organism_171 | Subdoligranulum unclassified | 1.27E-02 | 1.5560202207333 | 627 | 5 |
| Organism_152 | Roseburia hominis | 1.18E-02 | 1.7903389716571 | 1500 | 20 |
| Organism_138 | Coprococcus sp ART55 1 | 1.16E-02 | 2.2748823646463 | 701 | 8 |
| Organism_163 | Ruminococcaceae bacterium D16 | 1.12E-02 | -4.30319855302151 | 1390 | 30 |
| Organism_162 | Faecalibacterium prausnitzii | 9.80E-03 | -1.57257090414346 | 1220 | 18 |
| Organism_53 | Coprobacter fastidiosus | 9.67E-03 | 6.06805781620637 | 1503 | 19 |
| Organism_40 | Bacteroides massiliensis | 9.49E-03 | 1.53976594131914 | 1602 | 20 |
| Organism_136 | Coprococcus comes | 9.19E-03 | -1.68577511310286 | 116 | 1 |
| Organism_74 | Alistipes indistinctus | 8.46E-03 | 1.2651644466561 | 1447 | 19 |
| Organism_20 | Collinsella aerofaciens | 8.42E-03 | -1.82725111812987 | 1349 | 20 |
| Organism_123 | Eubacterium hallii | 7.81E-03 | 1.07627573371101 | 144 | 3 |

Robust taxa of the IBD dataset, identified by their internal identifier, as well as their current name. Taxa are classified by decreasing average Gini importance score, over all 200 RFs trained at the optimal selection level (20 per run, 10 runs). Extra information include: the ratio between the average abundances of the taxon in control and unhealthy profiles, ponderated by -1 if the taxon is most present in the unhealthy profiles, the amount of FAs attached to each taxon, and the number of robust annotations within them.

chemical products of the annotation and eventual alternative names of the annotation were similarly tested, followed by related (parent or child) annotations, and finally, the linked pathways listed in the BRENDA database [36]. From this exploration, the annotations were given a bibliographic relevance grade of 1 (most relevant to the disease) to 4 (least relevant to the disease) based on the following criteria:

Category 1: A direct link was established between the annotation, or a direct product metabolite, and IBD. This can come in the form of an explicit description of the metabolic mechanism's involvement, or simply in the form of measured differential presence between unhealthy and control individuals. Note that conclusions derived from other ML-based approaches were not considered to be sufficient evidence, as they could suffer from biases similar to our approach.

Category 2: A direct link was established between a similar metabolic function and the disease. Were considered as similar: proteins or enzymes from the same family as the one involved in the annotation (i.e.: ATP-dependent and ATP-independent citrate lyases), and parent and child annotations, signaling notably that the annotation is indeed relevant, but at the wrong scale.

Category 3: An indirect correlation was established between the annotation and the disease. This can mean that the annotation was not directly linked to IBD, but that it is involved in a larger pathway or expressed by a taxon that has significance.

Category 4: No leads were found, or the annotation was proven to be irrelevant.

Among the robust annotations, several were found through bibliography to be relevant to the disease when expressed in the host organism as opposed to the microbiota. We considered both cases as a link found between the annotation and the disease, following the idea of permeability and interactions between the microbiota and its host [37].

When available, we also retrieved the group, namely unhealthy or control, most likely to express these annotations according to the bibliography. At the same time, SPARTA also retrieves the group that most expresses each of these robust FAs (see Materials and methods). We confirmed these associations between FA and group with limma as well, for better robustness. We found that bibliography predictions and prevalence in the IBD dataset patients were in agreement in 47% of cases. FAs where disagreement exists between bibliography and SPARTA/limma might point towards the rescue of important functions in the host by the microbiota [38].

A complementary comparative analysis was conducted through a $Chi^2$ contingency test with a 95% p-value threshold between the prevalences of each bibliographic category in the robust selection and those of randomly selected non-candidate annotations (see S2 Table for details). This test established that the robust group significantly diverged from the non-candidate group. This significant difference is notably driven, as seen in S2 Table, by a comparatively increased proportion of Category 1, and a decreased proportion of Category 4 annotations in the robust subset compared to the non-candidate selection. These results support the notion that SPARTA is a relevant selector of information.

**Exploring the pairings between robust taxa and annotations highlights their non-redundancy.**   The usage of the EsMeCaTa pipeline [10] in building functional scores allows us to make explicit and quantify the links between taxa and their FAs, as this tool retrieves the annotations associated with a given taxon in the UniProt database [24]. Applying the pipeline to the IBD dataset (443 taxa and 10,196 FAs), the results show that annotations can be associated with 47.8 taxa on average. One annotation is associated with the most taxa (437 taxa out of 443): GO:0016021, which is attached to the cellular membrane component and is therefore expected to be extremely widespread. Unique associations account for 37.5% of all annotations, thus a majority of annotations are associated with more than one taxon. Overall, no function is perfectly ubiquitous, and the majority of functions are linked to several different taxa.

To quantify functional redundancy among taxa, we used Jaccard proximity to measure the similarity of their functional associations. Taxa with a Jaccard proximity of 95% or more were considered functionally identical. Our analysis showed that 77.2% of the taxa do not have such close neighbors, indicating that they maintain distinct functional profiles from each other, despite sharing many annotations with other taxa. Detailed results are given in S5 File.

The observed disparities between taxonomic and functional profilings prompt the question of whether these profiles equally provide valid descriptions of a subject's microbiota. A potential drawback of the taxonomic scale is the cumulation effect, wherein individual taxa may have little significance but contribute significantly to an essential metabolic process when grouped. As a result, this collective impact might go unnoticed when focusing solely on individual taxa. The dynamics in terms of specificity between annotations and taxa are illustrated in Fig 5, which plots the amount of robust taxa associated with each annotation as a function of the total amount of associated taxa. For illustration purposes, the represented annotations

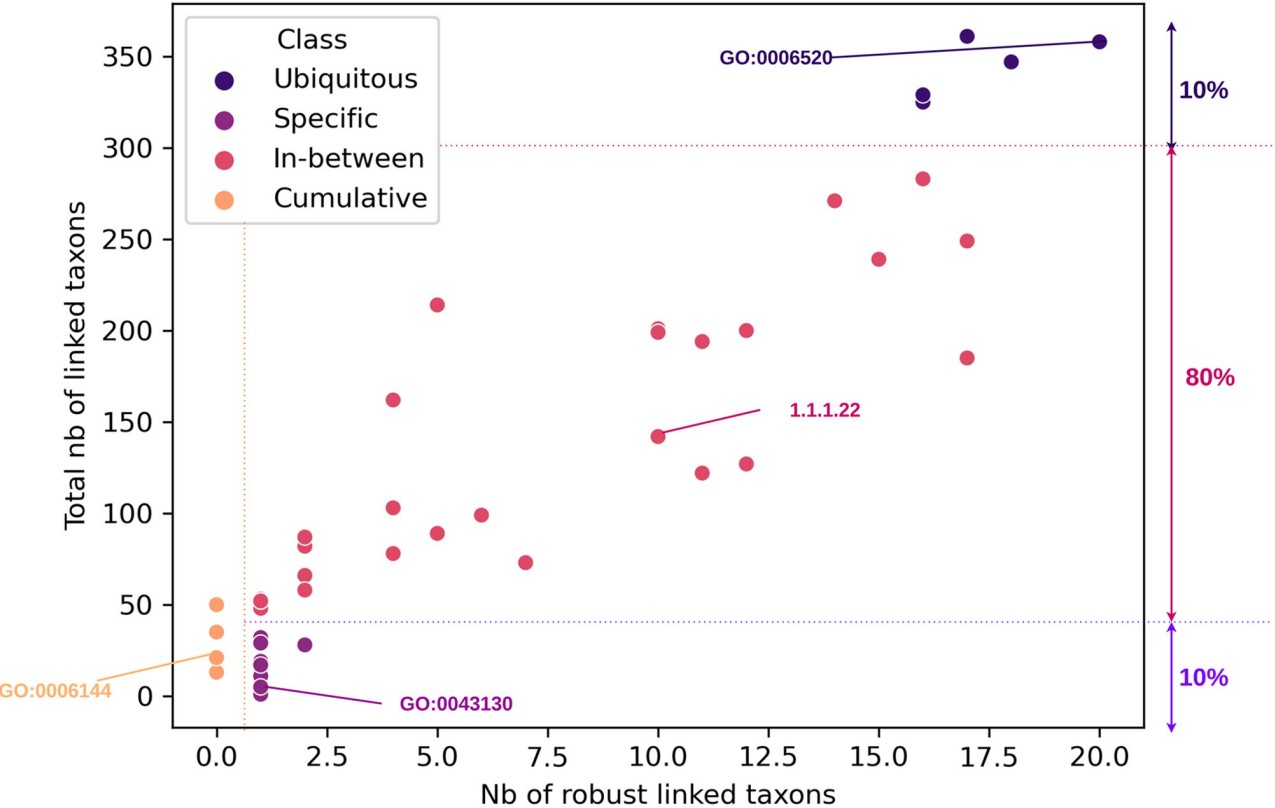

**Fig 5. Number of taxa associated to each robust annotation, as a function of the number of associated robust taxa for the IBD dataset.** Four groups of annotations are represented, three of which were determined based on the total amount of taxa attached to the annotation: those within the top 10% of these values' scale were labeled 'Ubiquitous', those in the bottom 10% were labeled 'Specific', and the others were labeled 'In-between'. The final category corresponds to the robust significant annotations with no relationship to the robust significant taxa ('Cumulative'). The highlighted annotations are those used as illustrative examples in Fig 6.

were assigned to four profiles based on their number of associated taxa. We labeled the top 10% as "Ubiquitous" (5 annotations, top right in Fig 5), the bottom 10% as 'Specific' (18 annotations, bottom left of Fig 5), and all others were labeled 'In-Between' (32 annotations). Finally, a fourth category was drawn up, independently of the previous criteria, containing 4 annotations that have no link to robust taxa, which we labeled as 'Cumulative'. This representation shows that important annotations have differing relationships to their taxon counterparts and that an annotation's importance can stem from the influence of several taxa, as is notably illustrated by the 'Cumulative' class.

A detailed illustration of annotations' pairings with their taxon counterparts, as well as the strength of these links determined as described in Materials and Methods, is proposed in Fig 6. The represented annotations were taken from each of the categories illustrated in Fig 5: GO:0006520 as representative of the 'Ubiquitous' class, 1.1.1.22 for the 'In-between' class, GO:0043130 for the 'Specific' class, and GO:0006144 as a 'Cumulative' example.

From top to bottom in Fig 6, the first annotation (GO:0043130) is a case in which the feature's significance appears to be due to a strong association to a single robust significant taxon, namely *Akkermansia muciniphilia*. This taxon has an established impact on IBD remission, and is researched as a potential probiotic treatment of the disease [39]. This is also in accordance with the annotation's differential expression between profiles, as seen in Table 5, where

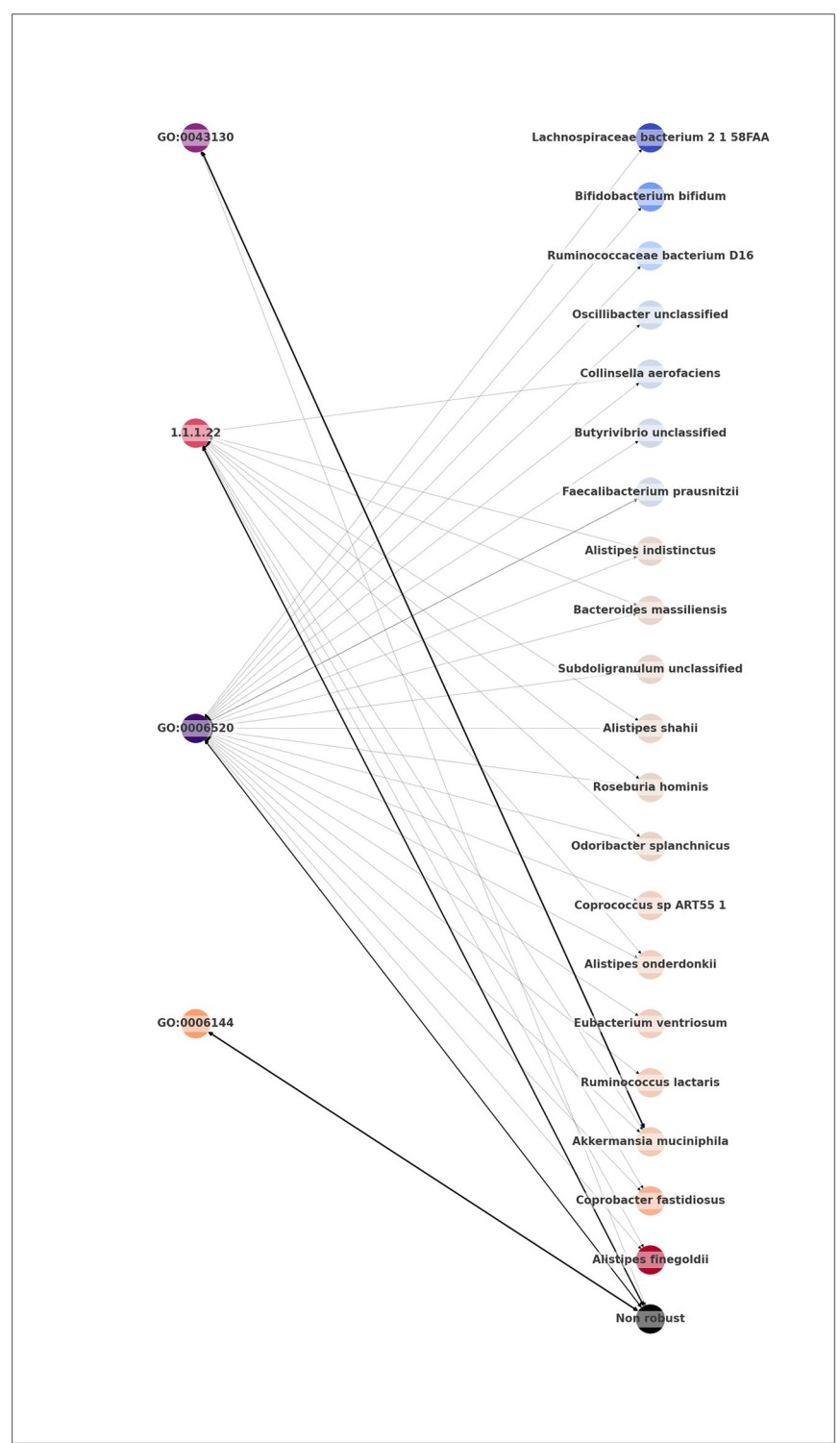

**Fig 6. Associations between robust functions and the associated robust taxa predicted by SPARTA, for the best iteration on the IBD dataset.** Depicted annotations were selected to be representative examples of the different categories highlighted in Fig 5, and are presented with the same color scheme. Taxa are colored based on their normalized average differential expression between Control (red) and Unhealthy (blue) profiles. The width of the connections is proportional to the importance of the association. The arrow between a given function and the generic 'Non-robust' node represents the contribution of non-robust taxa to the considered function.

**Table 5. Distribution of samples within the datasets of reference.**

| isease | Dataset | Total samples | Control samples | Patient samples |
|---|---|---|---|---|
| Liver Cirrhosis | Cirrhosis | 232 | 114 | 118 |
| Colorectal Cancer | Colorectal | 121 | 73 | 48 |
| Inflammatory Bowel Disease | IBD | 110 | 85 | 25 |
| Obesity | Obesity | 253 | 89 | 164 |
| Type 2 Diabetes | WT2D (European Women Cohort) | 96 | 43 | 53 |
| | T2D (Chinese Cohort) | 344 | 174 | 170 |

the annotation is shown to be expressed in the control samples almost 3 times as frequently on average as it is in the sick samples. This kind of relationship could either indicate that this 'Specific' annotation derives its importance in our predictions from its strong and specific attachment to an important taxon, or that its impact on the disease is an important factor to explain this taxon's beneficiary influence. GO:0043130 corresponds to ubiquitin binding, a mechanism that is known to regulate the inflammation process of intestines via different signaling pathways [40], and is categorized as a Category 1 annotation by our bibliographic research, showing that in the case of our example, the effects of the annotation and of its specifically associated robust taxon align. It should be noted that, as mentioned in our earlier discussion around our bibliographic work, the differential expression of a feature can be contradictory with its known effects, and should therefore be treated with caution. The second and third annotations (1.1.1.22 and GO:0006520), respectively from the 'In-between and 'Ubiquitous' groups, are very widespread among robust taxa, without any particularly strong link to any of them. In cases such as these, meaning metabolic functionalities commonly expressed within taxa, the issue of significance is shown to not be a purely binary question of expression or absence, as both annotations are consistently present in both unhealthy and control profiles. Finally, the last annotation (GO:0006144) is exclusively linked to non-robust taxa. All such annotations, from the 'Cumulative' group, are associated with several taxa (13 minimum), meaning that their importance results from the cumulated influence of multiple, individually non-significant taxa, that have a significant role when grouped functionally. The reverse associations, plotted in S7 Fig, show that this form of cumulation is specific to FAs: the robustly significant taxon with the least associations to robust significant annotations, *Coprococcus comes*, is still shown to have a non-zero amount of correlations to robust annotations.

## Discussion

Through the implementation of a new approach involving microbiota functional profiling, classification and variable selection, we have shown that the translation of the microbiota into functional profiles gives non-significantly different performances when compared to microbial profiles on 5 of 6 datasets. Through repetition, we also put forward a robust subset of discriminant variables. These selections were shown to be more reliable than those obtained by a state-of-the-art method, and their contents were validated through a manual bibliographic research on an example. The interconnections between selected taxa and functional annotations were also analyzed and revealed that important annotations emerge from the cumulated influence of non-selected taxa.

### From bacteria to functions

The first step of the SPARTA pipeline involves predicting annotations for the input taxonomic affiliations. To do so, we chose to rely on the EsMeCaTa pipeline, for the ease of its direct

application to microbial profiles, as well as the presentation of its outputs which records the inter-associations between taxonomic affiliations and FAs, making it more suitable for our subsequent analyses. Though our manipulations were made on data derived from MGS sequencing, EsMeCaTa is also capable of processing data derived from 16S sequencing.

It is however not the only tool available with the purpose of predicting functions, notably with the aforementioned PiCRUSt [8, 9] and HUMAnN [5–7] pipelines, which are widely exploited for 16S and MGS profiles, respectively. The cited works of Jones et al. [21] and Douglas et al. [20] notably rely on them. EsMeCaTa's exploitation also comes with caveats, as its reliance on UniProt means that any bias in the remote database would impact the tool as well, such as the inclusion of proteomes not adapted to the samples' environment of origin. The use of taxonomic profiles as an input makes the process lighter in terms of computational resources, but also makes the tool reliant on the quality of the preprocessing steps, as there is no referral to the original reads. Finally, it should be noted that EsMeCaTa, being reference-based, does not provide a quantification of the FAs within the sample itself, as SPARTA has to rely on its own manipulation based on EsMeCaTa's results to give an estimation of the abundance of expression of these annotations.

A comparison of results from our EsMeCaTa-based approach to those obtained from a profile obtained through HuMAnN3 [7], presented in S2 Fig, shows that both approaches give similar performances. However, processing patients samples with HuMAnN3 resulted in an over four-fold increase in terms of computation time, and required handling inputs of 442 GB, compared to EsMeCaTa's 302 kB entry (see S3 Table). Generally, it remains an open question to choose the right trade-off between computation time, classification performance, and interpretability when handling microbiota data. The modular implementation of SPARTA, allowing the user to directly specify functional profiles, aims at providing the corresponding flexibility to adjust the pipeline to the type of raw data (MGS or 16S data) or the phenotype of interest.

Finally, the scores calculated by SPARTA presented in this article are processed with the TF-IGM normalization [41], presented in Materials and Methods. This manipulation exacerbates the scores of the most differentially expressed annotations, heightening their highest scores, and lowering their lowest, to facilitate classification. A caveat of this approach however is that, as a cost for making the profiles more discriminating, it can enhance biases inherited from the database or from the taxonomic profiling.

## Comparative analysis of different approaches

**Microbial and functional profiles.**    A central discussion point of this study is the pros and cons of exploiting the microbiome's FA data as opposed to the explored taxonomic profiles for classification and interpretation. It should first be noted that there is an inherent bias to the exploitation of metagenomic data [42], notably concerning the taxa of lower abundance which are susceptible of being false positives.

In terms of classification performance, as shown in Fig 3 and discussed in Results, the taxonomic profiles remain a better overall predictor of disease state, though the difference is not significant in most cases. These results are in line with the findings of previous studies [20, 21], and can be explained by the increase in the amount of features contained in the functional profiles. Indeed, for a set amount of data, augmenting the number of variables past a certain point is known to be detrimental to model performance [43], and the switch to functional profiles comes with 22 times as many variables on average, without any additional samples to balance this. This hypothesis is further supported by the fact that variable selection increases the functional profiles' classification performances more consistently than the taxa.

The main benefit of the functional profiles is that they are more in line with the current demands of the medical community [4] when it comes to the required precision level for biological interpretation. A potential caveat however would be the optimal amount of features retained by SPARTA, which greatly varies between both profiles as seen in Table 1, with the amount of annotations retained for optimal classification being often greater than the equivalent for taxa. It seems intuitive that more metabolic functions would characterize unhealthy and control profiles when compared to taxa, however, the total amount of retained information in the case of FAs appears to be too extensive for biological interpretation to be practical in most of our examples. As such, we would recommend that interpretation of the FAs be limited on the first approach to the robust subset outputs, which are in more manageable numbers, though these lists are unlikely to extensively cover all of the features relevant to the characterization of the disease.

**SPARTA feature selection.** SPARTA exploits the feature importance rankings that are inherent to the RF method to perform a selection of variables. This selection step impacts classification performance, as discussed in Results, but is also important for the interpretability of the outputs, by highlighting the important elements within an otherwise overwhelmingly large list of features.

Our results highlight the need to perform at least one iteration and several repeated runs to reduce the dimensionality of the functional datasets, while maintaining the classification performance, and derive a list of robust FAs. The number of required iterations depends on both the dataset and the user needs in terms of classification performance and interpretability. Concerning the number of iterations to perform, in this article, we presented results obtained over 10 runs, comprising 5 iterative selections each, and implemented these values as default for SPARTA. These values were chosen as a compromise between execution time and the robustness of the results. S4 and S8 Figs illustrate the impact that a variation of these parameters can have on the results. When it comes to iterative selection, S4 Fig showcases that the first selection is always by far the most important, and there is little variation in selection sizes past the second selection. Therefore, 2 selections could also be perceived as the upper limit by some users, though some of our datasets have shown better classification performance beyond this level of selection. S7 Fig illustrates, in the case of the IBD dataset, that the sizes of both the functional and taxonomic Robust selections stabilize and hit a plateau after only a few runs. In both cases, 10 runs is sufficient to attain stable content for the Robust selection. This conclusion could however only be attained *a posteriori*, once the results had been obtained. A user may want to reduce the amount of runs operated by SPARTA but should bear in mind that these results may vary depending on the dataset. Generally speaking, SPARTA's criterion for optimal variable selection is to retain the subset that generates the best classification metric after one variable selection. An automatic test is implemented to ensure that classification performance after one iteration is not significantly lower than the one obtained with the initial dataset. Though this constitutes a strong basis for a first approach, previous works have also warned of it being potentially deceptive and encouraged to investigate the significance of the evolution in performance measurements [23]. As is, the exploration of the Confident subset or the exploitation of a lower level of selection than SPARTA's proposal could be envisioned by the user if the content included in the recommended Robust output is deemed insufficient. Similarly, a higher level of selection can be exploited if the proposed amount of Robust variables is still overwhelmingly large. Users should also be mindful that the output list may not be as relevant if the classification performances are low.

We also compared SPARTA's selection to other approaches, as reported notably in Fig 4 (A). By relying on an automatically computed cutoff threshold, our approach has proven to be more adaptative and robust than selections based on common fixed thresholds. The relevancy

of exploiting RFs to perform selection as opposed to a more direct statistical comparison of unhealthy and control profiles was also highlighted when SPARTA's selections are compared to those obtained with limma, which measures differential expression. While it proved itself to be an efficient selector on datasets with clear distinguishing features, the latter tool did not detect any candidate features at realistic adjusted p-value thresholds when applied to half of our test datasets and did not have the internal coherence to generate a robust shortlist in two thirds of them. SPARTA on the other hand provided a robust subset for all datasets, showing it to be more consistent than limma when it comes to variable selection, especially in complex problems.

RFs are known to be capable of finding non-linear solutions to a problem [44], which explains the fact that a large amount of the information highlighted by SPARTA, including within the robust subset, remains undetected by limma even when the p-value threshold is unrealistically high, as shown by the results of S5 Fig. As such, the content of SPARTA's selection includes new information when compared to what can be extracted from linear comparisons.

**Classification methods.**   Classification performances in the context of FAs have been reported to be on par or slightly inferior to classification performances based on taxa [20, 21]. This is also consistent with our observations. As a result, current FA-based approaches might not be best used for direct diagnostic prediction. The conditions in which a sample has been obtained, sequenced, and processed most likely impact classification performances, even for the same disease (see the differences in performance obtained on T2D and WT2D). The main advantage of current FA-based pipelines, SPARTA included, lies in the extraction of a robust list of important FAs, related to a dataset of interest, rather than the production of a ML model that is generic and directly reusable without need for retraining.

It should also be noted that in this article, only binary classification tasks were tackled. However, the key methods on which SPARTA relies are all compatible with multi-label classification tasks (SVM, RF, evaluation and importance metrics). As such, the pipeline could be compatible with such analyses.

## Post-processed outputs

**Output interpretability.**   SPARTA's end output is a shortlist of interconnected features, illustrated notably by the examples in Tables 3 and 4. The method emphasizes the selection's robustness, as it is derived from the consensus of several repetitions, and adaptability, as the threshold for selection is based on an automatic calculation rather than a fixed rank selection. Its content also underwent bibliographic validation, in the case of the IBD dataset's output. Though the list is likely not exhaustive, SPARTA's selection was shown to be significantly enriched in bibliographically significant features. This supports SPARTA's efficacy when it comes to highlighting factors that discriminate health profiles, though this should also be confirmed on the outputs obtained on other diseases.

The previously reported mismatches between the differential score-based profile attributions of SPARTA, which match those of limma, and the conclusions of bibliographic research show that, in all probability, the underlying biological mechanisms involving these pathways are complex enough that a simple differential association is not sufficient to predict if an annotation is beneficial or detrimental to host health in the context of a given disease. A compensation mechanism could also be at play, as the gut microbiota is known to have the potential to compensate for metabolic functions that are lacking in the host [38]. A finer analysis of the RF's trained decision trees could give more appropriate insight into this issue.

It should also be noted that several annotations couldn't be directly linked to IBD through bibliography (categories 3 and 4). These features deserve special attention, as they could be the result of a weakness of the method, or novel perspectives for research surrounding the disease.

The combination of SPARTA's outputs with a visualization method adapted for both of the employed nomenclatures, namely GO terms and EC numbers, would also be a complement to our outputs, allowing for a more intuitive exploration of their biological ramifications. A visualization such as this one could be the basis for an interpretation module for SPARTA.

**Exploring links between taxa and functions.**   Through Figs 5 and 6, we established the reality of a cumulation effect, with taxa that are less prevalent ending up having a detected influence on the microbiome's metabolism through their combined contribution to a functional niche. This observation further supports the importance of exploiting microbiota information at the functional level rather than at the taxonomic level. Annotation GO:0006144, which corresponds to the purine metabolic process and is represented in orange in Fig 6, is a good illustration of this approach's advantages. SPARTA's outputs show that this annotation was not correlated to any robust taxon, and therefore would be difficult to derive from a taxon-based approach. Indeed, the bibliography shows that this annotation was linked to IBD through oriented research following a first mechanistic study [45], where our approach was capable of identifying it efficiently and without any pre-orientation.

## Applicability of the SPARTA pipeline and perspectives

Though it was tested on gut microbiota data, this method's generic applicability can extend to other types of microbial communities. We focused on method robustness, presenting consolidated and exhaustive shortlists that showed agreement over 10 pipeline iterations without cherry-picking.

These first results present a proof of concept for highlighting differentiating features in biological datasets through Machine Learning-based classification and variable selection, and establishing that integrating inter-associated taxa and functions for disease state classification with the gut microbiota enhances interpretability and exposes a functional cumulation effect. It also presents opportunities for improvement. Method-wise, alternatives to the already implemented approaches could be envisioned, for example using other hyperparameter tuning methods (Bayesian Hyperparameter Optimization [46] to replace GridSearch for example) or tree-based approaches, such as XGBoost [47].

Integrating more specific external knowledge, such as individual clinical metadata, could enhance the interpretability of the questions for Machine Learning models. The integration of this information could also help classification, especially when they lead to a rapid and significant change in microbiota composition. For instance, the menstrual cycle [48], diet [49], or antibiotic treatment [50] could be recognized and accounted for by the models. To further the comprehensiveness of our outputs and filter potential redundancies within annotations, we could explore leveraging Semantic Web information surrounding GO terms and EC numbers to aggregate or expand the existing information from UniProt. This could be the subject of future work.

## Materials and methods

### Datasets

SPARTA was tested and benchmarked using publicly available species-level abundance profile datasets from the MetAML repository [19] and processed for DeepMicro [18], concerning subjects diagnosed with a variety of diseases: Cirrhosis [51], Colorectal Cancer [52], IBD [53], Obesity [54], and T2D on a Chinese [55] and a European [56] cohort. Each subject in these

datasets had their gut microbiota sampled and sequenced with whole-genome shotgun and Illumina paired-end sequencing. The results were processed by the authors of the MetAML and DeepMicro tools [18, 19] as per the standard procedure described by the Human Microbiome Project [57], then converted to species-level relative abundance profiles via the MetaPhlAn2 tool [58] with default parameters. Sub-species level features were then filtered using the MetAML tool [19].

Each cohort includes a portion of healthy control individuals, in addition to those who suffer from the disease in question. The proportions of each group in our cohorts are detailed in Table 5.

## The SPARTA pipeline: A Machine Learning-driven method for paired analysis of taxonomic assignations and FAs

An implementation of SPARTA in Python is available on github at https://github.com/baptisteruiz/SPARTA. The presented results were obtained from running in a Conda (version: 23.11.0) [59] environment that contains the EsMeCaTa pipeline (version 0.4.2) [10], as well as the following Python packages: pandas (version: 1.4.3) [60], numpy (version: 1.21.2) [61], scikit-learn (version: 1.1.1) [62], matplotlib (version: 3.5.1) [63], joblib (version: 1.1.0) [64], seaborn (version: 0.12.2) [65], progress (version: 1.6) [66], goatools (version: 1.2.3) [67], Biopython (version: 1.79) [68], requests (version: 2.28.1) [69], kneebow (version: 1.0.1) [70] and SHAP (version 0.46.0) [27].

The pipeline can be launched following two steps. The first can be called with the `sparta esmecata` command, represented in Fig 7. This command takes as input a taxonomic abundance table and launches a run of the EsMeCaTa pipeline [10], preceded by formatting steps for the creation and formalization of EsMeCaTa's input from the given data. This step exploits the pipeline as described in a following section. This is followed by the calculation of the scores of the FAs obtained this way, following the method described further down and using the list of associations between taxa and annotations, as well as the original microbial abundances. This step can also involve data treatment, per the arguments parsed in the command line. For example, the taxonomic abundance profile can be forcefully converted to a relative abundance profile, with each value being recalculated as a percentage of the sample's total before the functional profile is calculated. Once we have the functional profile, its values can also be scaled, either using sklearn's [62] `StandardScaler` or TF-IGM, as described further in the Methods section, depending on the user's input.

The second part of the pipeline can be called with the `sparta classification` command. It takes as input a file containing the labels associated with each sample in the dataset, a functional and taxonomic description of the samples, a description of each taxon's affiliation, and a table indicating the occurrence of functions in each organism. It is possible to only give the functional table as input, in which case none of the latter three inputs would be required. The functional table given as input can be derived from `sparta esmecata` or can be calculated using another tool of the user's preference. This step involves the training of 20 successive RF classifiers to sort individuals according to their associated labels (i.e.: 'sick' or 'control'), based on the relative abundance profiles of their microbiota or their calculated mechanistic representation.

Once per run, before any training, a subsample of the full dataset, is set aside as a test set. This set can be determined through the use of sklearn's [62] `test train split` function, or the user can also specify their own, pre-conceived datasets. During training, the remaining data is randomly split into a training set and a validation set, with a respective 80% / 20% distribution. To account for the disparity in representation between the unhealthy and control

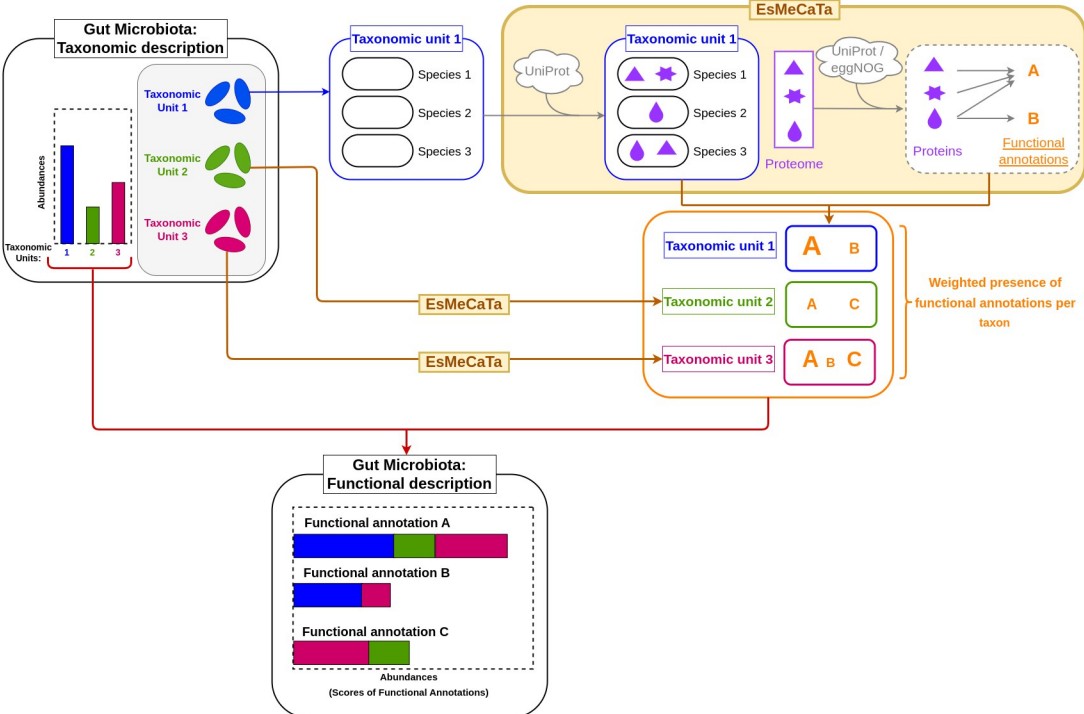

**Fig 7. Application of EsMeCaTa and calculation of Scores of FAs in the context of the `sparta esmecata` step of the pipeline.** The inputs represented here are taxonomic units, potentially containing several species. EsMeCaTa is compatible with this paradigm, but can also process data directly on the species level. EsMeCaTa interrogates the UniProt database to gather the proteomes of all species included in the input taxon. A meta-proteome for the entire taxon is then calculated, based on clustering using Mmseqs2 [71] followed by retention of clusters with a 95% incidence in all proteomes. UniProt is then interrogated a second time to retrieve the FAs of all of the kept protein clusters. A weighted association between taxon and annotation can be established in this manner. By combining this information with the taxon's initial abundance, a quantification of the FAs' expression can be measured.

individuals within the datasets, all classes were given weights proportional to their frequency, as implemented by scikit-learn's 'balanced' class weight parameter [62]. The training involves a Grid Search, as implemented by scikit-learn [62], to optimize the estimator's parameters in terms of the number of estimators per forest, the number of leaves per estimator, and the amount of information to which each tree has access. The split quality criterion is measured via the Gini Impurity metric. Optimal models were selected by GridSearch based on an internally conducted 5-fold cross-validation. This step exports a list of each trained forest's features' Gini [16] or SHAP [27] importances depending on user input, as well as their classification performances (see Results) on the validation and test datasets. The best performing model on the validation set is also exported. The final step involves averaging all features' importance scores over 20 training iterations, and selecting which ones are 'Significant' through a cutoff at the significance threshold, calculated as described further on. The list of all features above the cutoff threshold, listed by decreasing importance, is then given as output.

The user can require more than 1 iteration of the process, in which case a subset of the original microbial and functional profile files is created containing only the 'Significant' data. In the case where a data treatment method was given as input ('scaling' or 'tf_igm'), this step will be re-applied to the subset. After this, the training and variable selection steps will be repeated as many times as demanded, using the same test and validation sets as the first iteration's forests.

The entire process will be repeated, with the same parameters, as many times as dictated by the user through the requested amount of runs. Each of these runs will have a new subset of test individuals, and the user may also request that only a specified subset of the input profiles' variables be taken into account for each run. For instance, it is possible to filter out variables according to their abundance or prevalence. Once all requested runs have been completed, the shortlists obtained by all runs for each iteration are combined to categorize taxa and annotations as 'Robust' (outlined as significant by all predictors for a given iteration), 'Confident' (outlined as significant by at least 75% of predictors for a given iteration) or 'Candidate' (outlined as significant by at least one predictor for a given iteration). If the best obtained median RF AUC is inferior to 0.6, a message warning that the selection may be unreliable will be passed to the user.

The pipeline's main outputs are: the calculated functional profile, in the form of a csv table, the classification performances obtained by the pipeline through a graphical representation of the AUCs obtained at the best iteration level, and the Robust, Confident, and Candidate selections obtained for each iteration level in csv files. The details of the classification performances and variable selections per run and iteration are also made available to the user.

SPARTA's implementation also allows the user to classify the input data using a SVM [72] model instead of RFs. This option will however not proceed with variable selection, and can therefore only be used in single-iteration runs focused on performance. SVM parameters are also optimized through GridSearch, notably the regularization parameter, which tunes the impact of the loss function during training, and the classifier's kernel, which can be linear or Gaussian with Radial Basis (RBF), with a tuning of the gamma parameter (radius of each sample's area of influence) in the latter case.

## Shifting representations, from microbial to functional profiles

The first step of SPARTA's process is to transition from a representation of the microbiota on the scale of taxonomic affiliations to that of biological functions, by calculating the scores of the FAs linked to the input's taxa. In parallel, we are aiming to conserve the information linking together taxa and annotations, to expand upon this information later on. We also used the normalization of the annotation scores to introduce an *a priori* bias to boost the profiles of the best differentiating variables, in anticipation of the following classifying tasks.

**Associating FAs to taxonomic affiliations: The EsMeCaTa pipeline.**   The EsMeCaTa pipeline follows three steps. The first step, 'proteomes', takes as input a tabular that associates a given name for all the studied bacteria to their exact taxonomy. From this, EsMeCaTa interrogates the UniProt database for proteomes associated with the taxon in question. If none can be found, the step is re-iterated with the superior taxonomic rank, until at least one proteome can be associated with the unit. If more than 99 proteomes are associated with a taxon, a random selection of around 99 proteomes will be made, with respect to the taxonomic diversity of the initial proteomes set. The selected proteomes are then downloaded from UniProt.

The second step, 'clustering', selects protein clusters that are representative of the taxonomic unit within the downloaded proteomes. To do so, the MMseqs2 tool [71] is used to create clusters of similar proteins from the proteomes. If a protein cluster contains similar proteins from 95% of the proteomes attributed to the taxonomic unit, it will be retained as part of its meta-proteome.

The final step, 'annotation', fetches the FAs (GO terms and EC numbers) of the retained protein clusters by interrogating the UniProt databases. The final output is an ensemble of tabulars, one per taxonomic affiliation in the input, that contains all of the protein clusters kept in the taxon's meta-proteome and their FAs.

**Calculating a functional representation of the patient's microbiota from taxon-annotation pairings.** To compute a representation of the gut microbiota on the scale of the FAs, mixing information concerning its specific composition with the associated metabolic mechanisms, we give each annotation (F) a score, labeled as a Score of Functional Annotation (SoFA), within a subject sample (i), similarly to [8], according to the following formula:

$$SoFA_{F,i} = \sum_t n_{t,i} \times x_{F,t} \tag{1}$$

where $n_{t,i}$ is the abundance value of taxon t within sample i, and $x_{F,t}$ is the number of proteins within taxon t's proteome that are linked to the function F.

As such, each annotation's SoFA is equal to the sum of the abundances of all taxa that express it, weighted by the strength of said expressions, as measured by EsMeCaTa [10].

**Normalizing and scaling data based on expected relevance with TF-IGM.** The TF-IGM method [41] is used to normalize the results presented in this article. It was originally exploited in Natural Language Processing, as a method to highlight terms in a corpus of texts that are significantly present within a text while penalizing those that are too widespread. The formula had to be re-adapted to fit our data and circumstances, and in our pipeline, it is calculated based on the following two components:

- TF (Term Frequency): equivalent to the frequency of an annotation within the totality of a sample i:

$$tf_{f,i} = \frac{SoFA_{f,i}}{\sum_{j \in J} SoFA_{j,i}} \tag{2}$$

where $SoFA_{f,i}$ is annotation f's score within sample i, and J is the ensemble of the annotations recorded within sample i.

- IGM (Inverse Gravity Moment): for each annotation f, the calculated values for $tf_{f,i}$ are ranked in decreasing order and noted as $T(f)_1, \ldots, T(f)_n$, so that $T(f)_1 > T(f)_2 > \ldots > T(f)_n$, n being the total number of samples. We then have:

$$igm(f) = \frac{T(f)_1}{\sum_{r=1}^n T(f)_r \times r} \tag{3}$$

where r is the rank of the T(f) score in the previously defined order.

The total TF-IGM score of an annotation f within a sample i will then be:

$$tf\_igm(f, i) = \sqrt{tf_{f,i}} \times (1 + \lambda \times igm(f)) \tag{4}$$

where $\lambda$ is a value between 5 and 9. As per Chin et al.'s [41] recommendation, its value was set to 7 by default.

## SPARTA characterizes sample profiles with variables highlighted based on a non-linear approach

Having established two types of profiling for microbiotas, we then explore their potential in differentiating classes, such as individuals based on their health status. To account for the complex interdependencies of biological pathways in impacting host health, we relied on ML classifiers rather than linear statistical approaches to establish the relevance of variables when it comes to distinguishing unhealthy individuals from controls. A method for robust selection is also proposed here, with an automated shortlisting of variables based on their importance, and

a compilation of results accounting for consensus across multiple iterations of the method. The results presented in this article were obtained using 10 CPUs, and 100 GB of memory. Benchmarks of the classification process are available in S4 Table and show that SPARTA's execution time is linearly dependent on the number of requested runs.

**Training of RF models.** A RF [16] classifier is trained to sort individuals in two classes (here, patients or controls), based on the relative abundance profiles of their microbiota or their calculated mechanistic representation. Before any training, a subsample of 20% the size of the full dataset is set aside as a test set. During training, the remaining data is randomly split into a training set and a validation set, with a respective 80% / 20% distribution. To account for the disparity in representation between the unhealthy and control individuals within the datasets, both classes were given weights proportional to their frequency, as implemented by scikit-learn's 'balanced' class weight parameter [62]. Therefore, SPARTA differs from DeepMicro [18] by iterating the variable selection process: it introduces a test set—to evaluate the final performance of the model—and validation sets—to compute the performance of the RFs and derive the variable ranking for selection.

When measuring the performance of our classification algorithms, the metrics used were the Area Under the Receiver Operating Characteristic Curve (AUC) [73] averaged over 20 training iterations. All of the described operations related to the selection of test and validation sets, and the training of RF classifiers in the context of a GridSearch algorithm, are seeded to ensure reproducibility. The initial seed can be changed at the user's discretion.

**Extracting significant information from trained classifiers.** Following the classifier's training, the resulting feature importances are extracted. These importances can be based on one of two metrics, depending on the user's input. The first option is the Gini Importance metric, calculating the mean accumulation of the impurity decrease within each tree, as implemented in the Scikit-learn Python library [62]. The other option is the SHAP importance [27], which calculates each variable's contribution to a decision from the basis of a trained classifier. In our case, dealing with RFs, we relied on the SHAP package's [27] implementation of the TreeExplainer [74], which is an algorithm for the calculation of SHAP values optimized for RF models. If multiple iterations of the classifier's training are made, the feature importances are averaged over all iterations. Features are then ranked based on this metric in decreasing order. In SHAP's case, this ranking is made based on the absolute value of the importance scores.

Once ordered, we aim to distinguish a separation between the features that were essential to the clasifier's functionality, and those with a lesser impact. We place this threshold at the inflection point of the curve representing the decreasing importance scores, determined via an implementation of the Kneebow method [70], with all features above this point being labeled as "Significant", and those below as "Non Significant".

This process is iterated 5 times by default by SPARTA, and the optimal level of selection that is retained is the one that yields the highest median AUC during the classification process over 10 iterations of the pipeline.

**Repetition of the SPARTA pipeline.** To obtain robust results, the process of selecting a test subset, training classifiers, and extracting significant features for a set amount of iterations, was repeated over 10 runs in our manipulations. Variations in the training conditions, with different test subsets selected for each run, result in 10 different shortlists of significant features per iteration. We label as 'Robust' the features that constitute the intersection of these shortlists, as 'Confident' those that are present in 75% or more of them, and as 'Candidate' those that are present in at least one of them. The amount of times a variable is labeled as significant by the optimal level of selection is an indicator of how reliable it is for the distinction of the differentiated profiles.

## SPARTA lists and quantifies the pairings between significant variables

Beyond significant shortlists, SPARTA also aims to illustrate the links between taxa and FAs. EsMeCaTa's outputs list all of the annotations all of the annotations estimated to be expressed by each taxonomic affiliation in the database, as discussed in a previous Methods section. From this, we can establish the reciprocal association, linking all annotations to the taxa that express them. To quantify the reciprocal impact of a taxon on an annotation's score, we can calculate the following score:

$$\frac{\bar{n}_{M,i} \times x_{F,M}}{\sum_{M \in A(M)} \bar{n}_{M,i} \times x_{F,M}}$$

where $x_{F,M_x}$ is the number of proteins within taxon x's proteome that are linked to the function F, $\bar{n}_{M,i}$ is the average of the abundances of a taxonomic affiliation within a dataset and A(M) is the ensemble of all taxa associated with the annotation.

## Assigning a feature to a profile

SPARTA also involves associating taxonomic affiliations and FAs to either the unhealthy or control categories. To do so, the profiles (relative abundances for taxa, scores of FAs for annotations) of all individuals within the same category were averaged, and the features were associated with the profile where they were most prevalent on average.

## Application of HuMAnN3 to the IBD dataset

A functional profile was built from the raw reads of the IBD dataset, using the HuMAnN3 tool, in the context of a comparative evaluation of applications of SPARTA's classification approach to functional profiles from different sources. The process was conducted on the reads sequenced by Qin et al. [53], available at the European Bioinformatics Institute (EBI) website with accession code ERA000116. During the process, sample V1.UC-19 could not be processed properly, resulting in a functional table devoid of this sample. As such, in S2 Fig, the performances obtained on this profile were compared to classification performances obtained by applying SPARTA's functional profiling method to the IBD profile without the sample in question.

## Supporting information

**S1 File. Detailed classification performances per dataset, SPARTA run, and selection iteration.** The first sheet contains the detailed information as plotted in Fig 3: the average AUC scores, per run, for the overall best iteration level, for each dataset (taxa and annotations). The median of the average values, and p-values of the Mann-Whitney U-test comparisons between the taxon and FA average scores per disease are also given. P-values under the 0.05 threshold are considered significant and are highlighted with a *. Other sheets contain the details of each RF trained per run and iteration for each dataset (read: [dataset]_R_[run number]_It_[iteration number], with iteration numbers initialized at 0). The information given per sheet is: for each of the 20 RFs trained in this iteration and run, the optimal parameters found through GridSearch, the optimal threshold for probability prediction, and the AUCs on the training, validation, and test subsets.
(ZIP)

**S2 File. Detailed robust and candidate FA shortlists per dataset.** Each annotation is identified by its GO term or EC number, as well as its name. The complementary information given

includes: the annotation's average Gini importance over all RF models ('Average_importance'), the list of all taxa associated to the annotation ('Linked_taxa') and the sublist of robustly significant taxa within them ('Significant_linked_taxa'), and the profile it is associated to ('Family') supported by the average scores of the annotation in the patient and control samples. Outside of the robust shortlists, the number of SPARTA iterations that deem the annotation significant is also given ('Count').
(XLSX)

**S3 File. Detailed robust and candidate taxon shortlists per dataset.** Each taxon is identified by its internal identifier ('ID'), as well as its full taxonomy. The complementary information given includes: the taxon's average Gini importance over all RF models ('Average_importance'), the list of all annotations associated to the taxon ('Linked_Reactions') and the sublist of robustly significant annotations within them ('Significant_linked_Reactions'), and the profile it is associated to ('Family') supported by the average abundances of the taxon in the patient and control samples. Outside of the robust shortlists, the number of SPARTA iterations that deem the taxon significant is also given ('Count').
(XLSX)

**S4 File. Bibliographic exploration of the IBD dataset's shortlists.** The detailed conclusions of the bibliographic research on IBD's whole robust output, as well as random selections of 20 annotations that were non-candidates, and significant in 50% of SPARTA's runs. Bibliographic categories are as presented in Results. The categorizations are justified by quoted sources.
(XLSX)

**S5 File. Details of the pairwise Jaccard distance measurements between taxa based on their associated annotations.** Pairwise Jaccard distances between taxa, calculated based on their functional profiles as detailed in the 'Detail of taxon to annot links' sheet. The final column, 'Sum of close neighbors', counts the number of taxa with a distance of 0.05 or less from the one concerned. A value of 1 in this column means that the taxon in question only has itself for a neighbor.
(XLSX)

**S1 Table. Evolution of the average median AUC scores per dataset, on the validation and test sets, at increasing levels of variable selection, for taxonomic and functional (SoFA) profiles.** The top-performing selection levels on the test sets are highlighted in bold.
(PNG)

**S2 Table. Counts of the different bibliographic categories per researched selection, and p-values of a Chi$^2$ contingency test compared to the robust subset.**
(PNG)

**S3 Table. Comparative benchmarks of the execution time, sizes of input and output in applying HuMAnN3 and EsMeCaTa to the IBD dataset, with 10 CPUs and 150 GB of memory.**
(PNG)

**S4 Table. Execution time benchmarks for the SPARTA classification runs executed in the context of this study, with selection based on Gini and SHAP.**
(PNG)

**S1 Fig. Classification performances obtained with SPARTA on all datasets, using RF-based selections based on Gini and SHAP, and using SVM classifiers on the full dataset and the best-performing selection in terms of classification for Gini-based RFs.**

Performances at the top were obtained on the taxonomic profiles, those at the bottom were obtained on functional profiles obtained via EsMeCaTa. Similarly to Fig 2, the represented performances for the SPARTA (Gini, green for taxonomic and purple for functional, and SHAP, red) classifications are the median classification performances (AUC) for all types of profiles and each dataset, at the optimal level of selection over 10 full runs of the pipeline. SVM performances were obtained over a single run and were applied to the entire dataset (orange) or to the variable selections that correspond to the best performances for SPARTA Gini (blue). Performances obtained with SPARTA SHAP and SVMs were compared to those obtained with SPARTA Gini with a Mann-Whitney U-test. Those marked with a * showed a significant difference in distribution (p-value < 0.05). Consistent test and validation sets were used between all profiles for the classification tasks.
(PNG)

**S2 Fig. Classification performances obtained on the IBD dataset (minus sample V1.UC-19), annotated with EsMeCaTa (orange) and HuMAnN3 (blue), as well as on the taxonomic dataset (green).** Consistent test and validation sets were used for between all profiles for the classification tasks.
(PNG)

**S3 Fig. Sizes of the Robust, Confident, and Candidate selections obtained on each dataset over 5 iterations of SPARTA, using Gini and SHAP.** Top left: functional selections with Gini. Top right: functional selections with SHAP. Bottom left: taxonomic selections with Gini. Bottom right: taxonomic selections with SHAP.
(PNG)

**S4 Fig. Sizes and similarity of the individual Gini-based and SHAP-based SPARTA selections.** Top: sizes of the functional and taxonomic selections obtained by SPARTA with Gini and SHAP over 10 runs with 5 selective iterations, for all datasets. Bottom: similarity percentage between the individual Gini and SHAP selections, for functional and taxonomic profiles.
(PNG)

**S5 Fig. Robust, confident, and candidate shortlist overlaps for SPARTA and limma selections of comparable sizes on the T2D dataset.** The limma subsets were obtained with an adjusted p-value threshold of 0.255, chosen to obtain comparably sized candidate sublists between SPARTA and limma.
(PNG)

**S6 Fig. Classification performances obtained on the functional T2D and Cirrhosis datasets, selected by SPARTA (best performing selection) and by limma (alpha = 0.05).**
(PNG)

**S7 Fig. Associations between robust taxa and the associated robust functions predicted by SPARTA, for the best iteration on the IBD dataset.** Similarly to Fig 6, the color scale for the taxa is based on their differential expression between control and unhealthy profiles, and arrow width is proportional to the strength of the taxon's connection to the annotation. Relationships to non-robust annotations were not represented here for reasons pertaining to the readability of the figure. Represented taxa were chosen to showcase control and healthy representatives with high and low numbers of connections to robust annotations.
(PNG)

**S8 Fig. Sizes of the Robust selections obtained at each iteration level on the IBD dataset.**
(PNG)

## Acknowledgments

We would like to thank the GenOuest platform, which provided the computing resources used for obtaining the presented results. We would also like to thank Pauline Girard, Jeanne Got and Olivier Dameron for discussion and comments on the development of the method.

## Author Contributions

**Conceptualization:** Baptiste Ruiz, Anne Siegel, Yann Le Cunff.

**Data curation:** Baptiste Ruiz, Arnaud Belcour.

**Formal analysis:** Baptiste Ruiz, Anne Siegel, Yann Le Cunff.

**Funding acquisition:** Sylvie Buffet-Bataillon, Isabelle Le Huërou-Luron, Anne Siegel, Yann Le Cunff.

**Investigation:** Baptiste Ruiz.

**Methodology:** Baptiste Ruiz, Arnaud Belcour, Samuel Blanquart, Sylvie Buffet-Bataillon, Isabelle Le Huërou-Luron, Anne Siegel, Yann Le Cunff.

**Software:** Baptiste Ruiz, Arnaud Belcour.

**Supervision:** Anne Siegel, Yann Le Cunff.

**Validation:** Baptiste Ruiz, Yann Le Cunff.

**Visualization:** Baptiste Ruiz, Sylvie Buffet-Bataillon, Isabelle Le Huërou-Luron, Anne Siegel, Yann Le Cunff.

**Writing – original draft:** Baptiste Ruiz, Anne Siegel, Yann Le Cunff.

**Writing – review & editing:** Baptiste Ruiz, Arnaud Belcour, Samuel Blanquart, Sylvie Buffet-Bataillon, Isabelle Le Huërou-Luron, Anne Siegel, Yann Le Cunff.

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
