## [Decision Letter · Decision Letter 0]

25 Jun 2024

Dear Dr Le Cunff,

Thank you very much for submitting your manuscript "SPARTA : Interpretable functional classification  of microbiomes and detection of hidden cumulative effects." for consideration at PLOS Computational Biology.

As with all papers reviewed by the journal, your manuscript was reviewed by members of the editorial board and by several independent reviewers. In light of the reviews (below this email), we would like to invite the resubmission of a significantly-revised version that takes into account the reviewers' comments.

Baptiste et al. SPARTA : Interpretable functional classification of microbiomes and detection of hidden cumulative effects The manuscript entitled" aims to provide an alternative approach to profiling enriched functions in microbiome studies using metabarcoding. The author proposes a machine learning strategy facilitated by a random forest approach. One of the main challenges in profiling functions for a given microbial community—especially without specific questions on microbial ecology or biology—is managing the trade-off between too much or too general informativeness, as highlighted by reviewer #2 in their comments on Picrust and Human. This study holds potential to offer a feasible solution to this challenge, particularly for researchers without a strong microbiology background.

Four reviewers provided constructive feedback, leading to suggestions for significant revisions. Key points include:

1. GitHub Page Improvements: Enhancements to the GitHub page to improve accessibility and usability.

2. Additional Features for the Random Forest Model: incorporating more robust features into the Random Forest model.

3. Comparison with DEseq2: rationals of comparisons with DEseq2 need to be better addressed.

4. Statistical Analyses: Additional statistical analyses were also suggested.

5. Introduction: The introduction needs substantial revision to improve clarity and coherence.

6. Dataset Size and Runtime: More information regarding the size of the dataset and the time required to run SPARTA should be provided.

7. Moreover, the approach could become more useful if additional steps are included to statistically infer the dynamics of microbial communities and to profile the dynamics of microbial functions.

Overall, the manuscript requires major revisions before it can be considered for publication.

We cannot make any decision about publication until we have seen the revised manuscript and your response to the reviewers' comments. Your revised manuscript is also likely to be sent to reviewers for further evaluation.

Sincerely,

Zheng Wang

Guest Editor

PLOS Computational Biology

Stacey Finley

Section Editor

PLOS Computational Biology

Baptiste et al. SPARTA : Interpretable functional classification of microbiomes and detection of hidden cumulative effects The manuscript entitled" aims to provide an alternative approach to profiling enriched functions in microbiome studies using metabarcoding. The author proposes a machine learning strategy facilitated by a random forest approach. One of the main challenges in profiling functions for a given microbial community—especially without specific questions on microbial ecology or biology—is managing the trade-off between too much or too general informativeness, as highlighted by reviewer #2 in their comments on Picrust and Human. This study holds potential to offer a feasible solution to this challenge, particularly for researchers without a strong microbiology background.

Four reviewers provided constructive feedback, leading to suggestions for significant revisions. Key points include:

1. GitHub Page Improvements: Enhancements to the GitHub page to improve accessibility and usability.

2. Additional Features for the Random Forest Model: incorporating more robust features into the Random Forest model.

3. Comparison with DEseq2: rationals of comparisons with DEseq2 need to be better addressed.

4. Statistical Analyses: Additional statistical analyses were also suggested.

5. Introduction: The introduction needs substantial revision to improve clarity and coherence.

6. Dataset Size and Runtime: More information regarding the size of the dataset and the time required to run SPARTA should be provided.

7. Moreover, the approach could become more useful if additional steps are included to statistically infer the dynamics of microbial communities and to profile the dynamics of microbial functions.

Overall, the manuscript requires major revisions before it can be considered for publication.

Reviewer's Responses to Questions

**Comments to the Authors:**

Reviewer #1: Baptiste et al. present a comprehensive approach for integrating functional annotation of the gut microbiota into an automatic classification process, thereby enhancing biological interpretability. The authors proposed to use the taxonomic composition data as well as functional annotations as variables for health status classification. This work underscores the significance of understanding the functional aspects of the gut microbiota for medical applications, making the pipeline a valuable tool for microbiota feature selection.

However, several questions remain to be addressed.

1. As a machine learning classifier, the SPARTA should be compared with other machine learning tools such as SVM, as mentioned in the manuscript, rather than with DEseq2. DEseq2, intended for detecting differences in features like gene expression or taxa between groups, is not a classifier. Moreover, the authors should provide more detail on how they ran DEseq2, as the tool prefers original read count data over normalized data like relative abundance. Running Deseq2 with relative abundance may not get the best performance.

2. The results do not show that the functional features improve classifier performance. In fact, the performance is reported to be "slightly inferior." This unexpected result may arise from the fact that the functional annotation is predicted from the taxonomy rather than directly derived from the sequencing data. In fact, the dataset that the authors used are all whole-genome shotgun data. The authors should consider comparing species-level taxa classifiers with gene-level functional classifiers derived from the sequence data to test if functional annotation performs better as a classifier. Additionally, comparing SPARTA's performance with classifiers built by the original papers could be interesting. I noticed that for the T2D dataset, the original paper achieved AUC=0.81 based on functional gene profile, higher than all SPARTA repeats. But for the cirrhosis dataset, the original paper with 15 gene markers only achieved AUC=91.8% in the training set and AUC=83.6% in the validation set, much lower than SPARTA results.

3. For all types of machine learning feature selection tools, understanding why certain features were selected is crucial. It is challenging to interpret the results, especially those from Random Forest. The authors should consider adding content on interpreting the results and possibly integrating data visualization into SPARTA to aid users in interpreting the results better.

4. In the SPARTA pipeline, the authors introduce the '−i' argument and '−r' argument. It would be beneficial for the authors to test how many iterations and repeated runs are necessary to achieve a robust result, where increased repeats do not introduce new features and the feature ranking remains unchanged.

5. Is the random sampling process controlled? To achieve a reproducible analysis, it’s better to have a random seed control. Or to make sure the result is robust.

Reviewer #2: The manuscript presents an interesting workflow to classify microbiota compositions and their functional profiles using a Random Forest approach.

## Introduction

The introduction should be completely rewritten, as it fails to set the stage for the pipeline as it’s presented in figure 1, and presents the biological domain of microbiome analysis with a perceived disconnection with mainstream methodologies and best practices. A major flaw is the ambiguity between metabarcoding (16S) and whole metagenome sequencing (WMS) approaches.

The literature review in the first few paragraphs feels somewhat scattered and could be tightened up to more clearly delineate previous work using taxonomic profiles versus functional profiles. The key limitations of purely taxonomic approaches should be explicitly stated upfront to better motivate the shift towards functional analyses. The general feeling is a disconnection between the introduction and the commonly used workflows and procedures, and it’s not very clear how the package deals with metabarcoding or whole metagenomics sequencing. The acronyms OTU and ASV are used in metabarcoding analyses.

The link with function is mentioned citing Picrust and Humann: it should be noted that Picrust is used with metabarcoding datasets (but it’s known for the very poor

performance, as it uses a taxonomic marker to infer the function, which is basically impossible), while Humann uses whole metagenome sequencing reads, with a reasonable rationale. Mixing the two makes no sense in real life, unless the goal is to release another exercise of Machine Learning applied to biological datasets.

## Package

The authors released some code in a GitHub repository (https://github.com/baptisteruiz/SPARTA). This is very far from currently accepted standards for sharing bioinformatics packages, and until these issues are fixed, this manuscript should not be accepted:

* Poor documentation: there is a general poor structure of the code and repository, to the point that there are markdown errors in the README.md. The code in `DMmodif_export_test_ver.py` is mostly commented out, generally suggesting this is a primitive development version.

* Lack of tests: there is no automated test suite

* Lack of packaging: the program should be released as a Python package.

* Lack of references: there is a directory with “Inputs”, that is not documented and there is no clear reference of the source of the input files. Referring to lack of documentation, some files (such as `Label_abundance_Cirrhosis.csv`) are a bare list of values, with no practical use in real life (if someone wanted to use the package, they would need to create a chaotic set of files, making the process error prone).

* Dependencies: in the paper the authors make a list of dependencies. It’s trivial to add a package to the BioConda channel of Anaconda/Mamba, allowing you to install a package and its dependencies with a single command.

## Methods

I would recommend reorganising the section.

* starting with the datasets,

* adding a figure describing the EsMeCaTa pipeline

* Removing the parameters used by the script, using the opportunity to refactor the README where they belong

* Describing the output files (format) as they do for the input files

Reviewer #3: The review was uploaded as an attachment.

Reviewer #4: The manuscript "SPARTA: Interpretable Functional Classification of Microbiomes and Detection of Hidden Cumulative Effects" by Ruiz et al. introduces SPARTA (Shifting Paradigms to Annotation Representation from Taxonomy to identify Archetypes) a computational pipeline that integrates functional annotations (FAs) into a machine learning classification and use it to classify several disease data sets. SPARTA main benefit is to keep information about the connections between taxons and annotations to be able to provide interpretability into a classification.

My main concern with the paper is that Operational Taxonomic Units (OTUs) consistently outperform functional annotations (FAs) in classification tasks. Even then, the paper promotes the use of functional profiles, arguing that "the innovative potential of functional profiles resides more in their prospective contribution to a biological understanding of the diseases' mechanisms than in their use for automatic classification" (line 200). However, the number of FAs required for optimal classification with SPARTA exceeds 500 (Figure 3A). The discussion subsection "Microbial and functional profiles" acknowledges this issue and suggests using only robust FAs. However, for the cirrhosis dataset, which shows the best classification performance, the robust subset of FAs still includes over 200 features. Given the large number of features, it seems difficult, if not impossible, to gain a clear understanding of the disease mechanisms, plus such a large number of features, makes it more challenging to provide bibliographic categorization, as done in the article for IBD. Therefore, I am not convinced that using FAs offers any advantage over OTUs.

A second concern is the significant variability in classification results, which suggests that SPARTA would only be useful if one already has large, well-classified datasets. In such a case, one could test if SPARTA provides good classification for a particular dataset and then use it for future classifications of the same disease. This limitation significantly narrows the applicability of the method. Moreover, as evidenced by the type 2 diabetes datasets, the results are highly dependent on the specific dataset used (WT2D vs. T2D in Figure 2). Moreover, since the main point of the article is disease classification, there needs to be a comparison with other classification methods besides DESeq2, such as the mentioned SVMs (line 546). How well does SPARTA perform compared to other published classification methods specific to each dataset used? This comparison is crucial to evaluate the true effectiveness and relevance of SPARTA in the context of existing methodologies.

One final point: my understanding is that gut microbiota can change significantly in just a matter of days due to factors such as the normal menstrual cycle (Schieren et al., Exp Clin Endocrinol Diabetes 2024) or dietary changes (David et al., Nature 2014), as well as other influences like infections and antibiotic use. Given that your datasets are based on single time points per individual, it is important to discuss how these rapid temporal changes might impact your results. Acknowledging and addressing the potential effects of such variability on the accuracy and reliability of your findings is crucial for a comprehensive understanding of the limitations and applicability of your study.

Here are some other minor corrections:

Line 66: Small typo, 'litterature'

Line 169: Define AUC, it is the first time using the abbreviation. Also define it in the legend for Fig 2.

Figure 3B: Include this table as a table and not a subfigure.

Line 259: space missing in "adjustedp-value"

Line 319: How is the IBD dataset an "average representative" of your results when it is the dataset with the second best classification performance? I would have said that T2D is more in the middle of the classifiers.

Paragraph starting in line 440: I can locate the annotations in Fig 6, but I am not sure how you are going about in order in the paragraph, you say "from top to bottom" but start with the bottom (0043), then say "second and third" but point to the top and second from top to bottom. Following the order here is confusing.

**Have the authors made all data and (if applicable) computational code underlying the findings in their manuscript fully available?**

Reviewer #1: Yes

Reviewer #2: Yes

Reviewer #3: Yes

Reviewer #4: Yes

PLOS authors have the option to publish the peer review history of their article (what does this mean?). If published, this will include your full peer review and any attached files.

Reviewer #1: No

Reviewer #2: No

Reviewer #3: **Yes: **Jonas Coelho Kasmanas

Reviewer #4: **Yes: **Jorge A. Alfaro-Murillo
---

## [Decision Letter · Decision Letter 1]

30 Sep 2024

Dear Dr Le Cunff,

Thank you very much for submitting your manuscript "SPARTA : Interpretable functional classification  of microbiomes and detection of hidden cumulative effects." for consideration at PLOS Computational Biology. As with all papers reviewed by the journal, your manuscript was reviewed by members of the editorial board and by several independent reviewers. The reviewers appreciated the attention to an important topic. Based on the reviews, we are likely to accept this manuscript for publication, providing that you modify the manuscript according to the review recommendations.

All three reviewers considered this revision was a significant improvement from the previous submission, and brought up some minor edits for your attention. I am satisfied how you addressed our comments and suggestions. Good job!

Sincerely,

Zheng Wang

Guest Editor

PLOS Computational Biology

Stacey Finley

Section Editor

PLOS Computational Biology

All three reviewers considered this revision was a significant improvement from the previous submission, and brought up some minor edits for your attention. I am satisfied how you addressed our comments and suggestions. Good job!

Reviewer's Responses to Questions

**Comments to the Authors:**

Reviewer #1: The authors have addressed all my questions, and I don't have further concerns.

Reviewer #3: I would like to thank the authors for their responses to the first round of reviews. Their efforts have significantly improved the manuscript. The SPARTA resource provides a valuable entry point for machine learning analysis for microbiologists. It is also an easy-to-use solution with a more robust approach than the default parameters for feature ranking for the microbiologist audience that is not familiar with it. The manuscript addressed the comments successfully, and the GitHub was significantly improved. Currently, I have a few minor reviews to enhance the clarity of your work further:

1 - Abstract: I suggest replacing "reliable" with "consistent" when describing your solution, as it more accurately reflects the nature of your results.

2 - Discussion: Please include a brief discussion on why the inclusion of additional information (e.g., FA in addition to the taxonomy) might reduce model performance. More specifically, a discussion regarding the bias of the ML methods and their relationship with the dimensionality and the sampling of the dataset. This is particularly important for the biologist audience unfamiliar with the limitations of machine learning analysis.

3 - User guidance: To enhance reproducibility, it would be helpful to provide examples of software that can generate the input data required by SPARTA. This information could be included in the GitHub repository. For FA profiles, it is already clear that esmecata should be used. Additionally, in the Github, mention the Python version used. I initially tried to install Sparta in a Python 3.6 environment, but your suggested installation method (pip install -e .) only works on modern iterations of Python.

4 - The manuscript focuses on binary classification. Does SPARTA only work in binary problems? If this is indeed the case, please briefly mention this limitation in the text and provide more detailed information in the GitHub repository.

5 - Terminology: When referring to "Scores for FA," please ensure that you do not use "abundance" to avoid potential confusion.

6 - SVM details: When discussing the use of Support Vector Machines (SVM), please specify the hyperparameters used, particularly the kernel function as this significantly changes the SVM behavior.

7 - Future perspectives: In the discussion of potential future work, consider mentioning:

- The inclusion of other fine-tuning approaches beyond GridSearch

- The possibility of implementing new algorithms, specifically tree-based implementations tuned for the datasets (e.g., XGBoost)

-The addition of an interpretability module to describe and visualize the features

8 - Terminology (L381 of the manuscripts without tracked changes): Please revise the use of the word "expressed" as this is a reference-based approach.

These minor revisions will further improve the clarity and completeness of the manuscript. Once these minor reviews are addressed, I recommend the publication

Reviewer #4: The article has significantly improved since the initial draft. I particularly appreciate the new text in the "Classification methods" subsection of the Discussion. The new content provides valuable clarity. However, I recommend adding at least one sentence in the abstract to summarize the key point made in this subsection.

One critical aspect that still requires clarification is the use of FAs for the classifier, specifically regarding the results from Figure 3. Are all candidate FAs used for this classification? If so, my original concern from the previous review remains: classification based on FAs is inferior to using taxa while the set of candidate FAs appears too large—except possibly for Cirrhosis—to provide clinicians with meaningful insights into disease characterization. Could you consider using only the Robust or at least the Confident subset of FAs for classification? If so, how does the performance compare?

Additional minor details:

- In Figure 1, change "OTU" to "taxa," consistent with the rest of the article.

- In lines 217, 219, 743, and 867, follow "This" or "second" with a noun to clarify what is being referenced.

- It would be helpful to open the Discussion with a brief interpretation of the key findings.

**Have the authors made all data and (if applicable) computational code underlying the findings in their manuscript fully available?**

Reviewer #1: None

Reviewer #3: Yes

Reviewer #4: Yes

PLOS authors have the option to publish the peer review history of their article (what does this mean?). If published, this will include your full peer review and any attached files.

Reviewer #1: No

Reviewer #3: No

Reviewer #4: **Yes: **Jorge Alfaro-Murillo

Figure Files:

Data Requirements:

Reproducibility:

References:

---

## [Editor Report · Decision Letter 2]

22 Oct 2024

Dear Dr Le Cunff,

We are pleased to inform you that your manuscript 'SPARTA : Interpretable functional classification  of microbiomes and detection of hidden cumulative effects.' has been provisionally accepted for publication in PLOS Computational Biology.

Best regards,

Zheng Wang

Guest Editor

PLOS Computational Biology

Stacey Finley

Section Editor

PLOS Computational Biology

Good work!

---

## [Editor Report · Acceptance letter]

12 Nov 2024

PCOMPBIOL-D-24-00411R2 

SPARTA : Interpretable functional classification  of microbiomes and detection of hidden cumulative effects.

Dear Dr Le Cunff,

I am pleased to inform you that your manuscript has been formally accepted for publication in PLOS Computational Biology. Your manuscript is now with our production department and you will be notified of the publication date in due course.

With kind regards,

Lilla Horvath
